# Full Event Particle-Level Unfolding
# with Variable-Length Latent Variational Diffusion

**Alexander Shmakov**[1*]**, Kevin Greif**[2†]**, Michael James Fenton**[2]**, Aishik Ghosh**[2,3]**, Pierre Baldi**[1] **and Daniel Whiteson**[2]

**1** Department of Computer Science, University of California, Irvine
**2** Department of Physics and Astronomy, University of California, Irvine
**3** Physics Division, Lawrence Berkeley National Laboratory

⋆ ashmakov@uci.edu , † kgreif@uci.edu

## Abstract

The measurements performed by particle physics experiments must account for the imperfect response of the detectors used to observe the interactions. One approach, *unfolding*, statistically adjusts the experimental data for detector effects. Recently, generative machine learning models have shown promise for performing unbinned unfolding in a high number of dimensions. However, all current generative approaches are limited to unfolding a fixed set of observables, making them unable to perform *full-event* unfolding in the variable dimensional environment of collider data. A novel modification to the variational latent diffusion model (VLD) approach to generative unfolding is presented, which allows for unfolding of high- and variable-dimensional feature spaces. The performance of this method is evaluated in the context of semi-leptonic $t\bar{t}$ production at the Large Hadron Collider.

# 1   Introduction

Particle interactions can reveal new particles and forces, and allow for measurements of the parameters of theories that explain their dynamics. The detectors that measure the final state particles produced by such interactions have limited resolution and efficiency, introducing *detector effects* that must be accounted for in any statistical inference, for example via a simulator [1]. However, high-fidelity simulations of modern detectors are not universally accessible and are extremely computationally expensive [2–5]. An alternative approach, *unfolding*, is to correct the observed data for the effect of the detectors. This allows for statistical inference without access to the expensive detector simulator, and enables easier comparison of data from different experiments or with new theory predictions.

Traditional unfolding techniques [6–8] simplify this challenging task by binning data in a set of pre-selected quantities. These binned unfolding techniques become unfeasible as the number of dimensions grows beyond one or two, due to the curse of dimensionality [9]. Recent advances in machine learning (ML) have enabled unbinned unfolding in many dimensions. These methods fall into two categories: discriminative methods that train classifiers to reweight synthetic data distributions to an estimate of the truth level distributions [10,11], and generative methods that model the distribution of a set of truth level observables given the corresponding detector level distributions [12–18]. Ideally, these methods would enable full-event unfolding, in which the kinematics of every final state particle are unfolded. The resulting unfolded datasets could be used to measure any observable without the need for expensive detector simulation, and even long after the experiment has been concluded. Since the number of particles in a final state can vary significantly due to inherently random processes, the ability to handle variable dimensionality events is a crucial element of true full-event unfolding. Discriminative methods have successfully performed high-dimensional (up to $\mathcal{O}(100)$ target observables) and variable-dimensional unbinned unfolding [19–22]. However, these methods may struggle in regions where the number of observed events is small. Because generative approaches require only synthetic data during training, they do not suffer from this limitation - however, handling variable dimensions can be a formidable challenge for existing generative unfolding methods [13,16].

Recently, Variational Latent Diffusion (VLD) models [18] were introduced and shown to be a powerful generative approach for unfolding a fixed number of observables. In this paper, the VLD approach to unfolding is extended to a variable dimensional set of observables, enabling

full-event unfolding even when the number of observed events is small. Full event unfolding with variable dimensionality is demonstrated for measuring top quark pair production, a common use-case of unfolding algorithms for experiments at the Large Hadron Collider.

One potential shortcoming of unfolding methods optimized only on simulations is the unfolded distribution's dependence on the prior distribution used to construct training data. If necessary, an iterative method first proposed in Ref. [14] can be applied to mitigate this dependence on the training set prior. This paper assess the prior dependence of a VLD model by evaluating it over an alternative testing set which contains shifts in the truth particle level distributions.

## 2  Background

Unfolding methods aim to sample from a truth distribution $f_{\text{truth}}(x)$, where *truth* refers to an unobserved state of interest to physicists. The observations include distortions introduced by the detector systems, and it is desirable to correct the measured data to remove these effects. Having only access to the observed detector-level data set $y = \{y_i\}$ with distribution $f_{\text{det}}(y)$, an unfolding method aims to learn the response function $p(y|x)$ which connects the two:

$$f_{\text{det}}(y) = \int dx \, p(y|x) f_{\text{truth}}(x) \tag{1}$$

The response function itself is unknown, but $(x, y)$ pairs can be produced through Monte-Carlo (MC) simulation. The truth distribution $f_{\text{truth}}(x)$ can be recovered via the corresponding inverse process if one has access to the posterior, a pseudo-inversion of the response function $p(x|y)$:

$$f_{\text{truth}}(x) = \int dy \, p(x|y) f_{\text{det}}(y) \tag{2}$$

The strategy of most generative unfolding methods is to build the posterior as a generative model trained on $(x, y)$ pairs[1], which can be used to sample from $p(x|y)$ and obtain the truth distribution via Equation 2. An important issue when choosing to directly model the posterior is that this quantity is itself dependent on the desired distribution $f_{\text{truth}}(x)$, the prior in Bayes' theorem:

$$p(x|y) = \frac{p(y|x) f_{\text{truth}}(x)}{f_{\text{det}}(y)} \tag{3}$$

Producing the sample of simulated data used to train the generative model requires choosing a specific $f_{\text{truth}}(x)$, which influences the learned posterior. In application to new datasets, this may lead to an unreliable estimate of the posterior density if the assumed prior is far enough from the truth distribution. A common method to overcome this challenge is to apply an iterative procedure, in which the assumed prior is re-weighted to match the approximation to the truth distribution provided by the unfolding algorithm [6]. Though application of this iterative procedure is not shown in this paper, the principle has been well-established for other generative unfolding methods [23], for which the conditions are similar.

In collider physics, there are multiple truth distributions of interest which could in principle be inferred from the detector level distribution. Ref. [18] applied VLD to *parton*-level unfolding; partons are intermediate particles produced directly from the hard scatter process but before the parton shower and hadronization processes [24]. For parton-level unfolding, the pseudo-inversion of the response function $p(x|y)$ then contains the pseudo-inversion of both the detector response and the parton shower and hadronization processes.

---

[1] A method for detector-simulation-free training is presented in Ref. [17].

In contrast to the parton showering and hadronization, the response of the detector to the final state particles from a collision event is very well modeled by simulators and generally independent of the hard-scatter process which produced the particles. Therefore it is usually preferable to unfold to *particle*-level: the set of stable particles directly before interaction with the detector. After identifying the stable leptons, the remaining stable particles can be clustered into *jets*. VLD is applied to particle-level unfolding in this paper, using the leptons and jets as targets. In contrast to the fixed dimensionality of parton-level unfolding, a particle-level event is of an inherently variable dimensionality since it must also describe jets that result from random processes such as initial- and final-state radiation. In this paper, VLD is modified to accommodate a variable dimension output, making it the first generative unfolding method with this capability.

## 3  Methods

Diffusion models are a class of generative models which have excelled in learning high-dimensional probability distributions at high fidelity. They have been been applied to a variety of generative tasks within high energy physics (HEP), including event generation [25–31], calorimeter shower simulation [32–39], anomaly detection [40–42], likelihood estimation [43], and unfolding [16,18] . They involve formulating a diffusion process in which samples from a data distribution are mapped to a pure noise distribution through addition of (usually Gaussian) noise, parameterized by a time step $t \in [0,1]$. In a standard formulation of the diffusion model [44,45], a neural network is trained to predict the added noise, conditioned on the original data sample and the time step $t$. The output of this model can then be used to reverse the diffusion process, allowing samples from the pure noise distribution to be mapped to a sample from the underlying distribution from which the training data is sampled. Many recent papers have improved the diffusion process, including a reinterpretation of the diffusion process as a stochastic differential flow [46], improvements in the solver used to generate samples from a learned flow [47], and moving the diffusion process into an abstract latent space [48].

Ref. [18] introduced variational latent diffusion models (VLD) which combine the interpretation of a diffusion model as a hierarchical sequence of variational autoencoders (VAEs) [49] from Ref. [50] with the latent diffusion [48] approach of operating the diffusion process within the learned latent space of a pre-trained VAE. The combination of these ideas allowed the construction of a conditional generative model in which both the VAE, which defines the latent space, and the diffusion model can be trained together in an end-to-end variational framework.

In this section, an extension of the VLD model is introduced, designed to unfold to a variable dimensional truth distribution, conditioned upon a variable dimensional detector-level distribution. Formally, the particle level unfolding problem is learning the distribution of a set of objects $X = \{x_1, x_2, \ldots, x_N\}$ conditioned on another set of objects $Y = \{y_1, y_2, \ldots, y_M\}$. It is crucial to note that the correspondences between entities in these sets are not strictly one-to-one and the cardinalities of these sets may differ: $M \neq N$. VLD is designed to learn the distribution $P(X|Y)$, which is conditional on all information in the set of objects $Y$ and captures the correlations between the individual elements of $X$. The latent space of the VAE is adapted to optimize the diffusion process, rather than being held fixed as the noise prediction network is trained. A diagram of the model is shown in Figure 1, and the individual components are described below.

Similarly to Ref. [18], the distributions are defined over a learned latent space $X = f_P(\mathcal{O}_P)$, derived from the original particle-level observations $\mathcal{O}_P$ with the help of a VAE. A similar mapping is learned for the conditioning set, derived from the detector-level observables with the help of a detector encoder $Y = f_D(\mathcal{O}_D)$. The VAE, detector encoder, and diffusion components

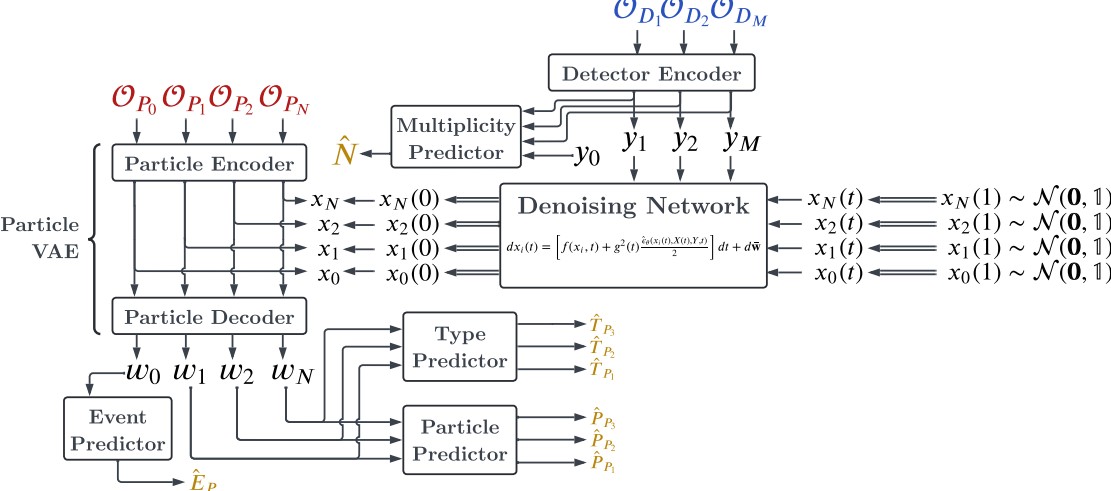

Figure 1: A Flow diagram of the components of Particle VLD. Pairs of particle level ($\mathcal{O}_P$) and detector level ($\mathcal{O}_D$) events are used to train the model using the loss functions introduced in Ref. [18]. At inference time, a detector level event is used to produce a multiplicity prediction and mapped to a latent embedding through the detector encoder. The multiplicity prediction and the latent representation of the detector level event are then used to condition the diffusion process, resulting in a sample from the latent space of the particle VAE. As with a standard VAE, the particle decoder is then applied to produce a sample from the learned conditional distribution $P(X|Y)$ and the particle encoder is not used in inference.

are trained as part of an end-to-end training scheme similar to Ref. [18], but with the addition of a multiplicity predictor, described below.

## 3.1 Particle VAE

The Particle VAE, consisting of an encoder and a decoder, is designed to learn an efficient mapping from the low-level observables of particle-level events into a latent space optimized for the diffusion process. Given the variable length nature of particle level events, a transformer [51] is used as the underlying architecture for both networks. Each event is comprised of observables associated with individual particle-level objects, along with global observables that encapsulate the entire event's properties. The encoder inputs include a set of $N$ particle-level objects, represented as $P_{P_i} \in \mathbb{R}^{D_{\text{Particle}}}$ for each object, complemented by a one-hot encoded vector $T_{P_i} \in \{0,1\}^{D_{\text{Type}}}$, which specifies the type of the particle object. Additional event-level observables are denoted as $E_P \in \mathbb{R}^{D_{\text{Event}}}$.

The set of observables for each particle-level object is denoted $\tilde{\mathcal{O}}_P = \{\mathcal{O}_{P_1}, \mathcal{O}_{P_2}, \ldots, \mathcal{O}_{P_N}\}$, where $\mathcal{O}_{P_i} = (P_{P_i} || T_{P_i})$ represents concatenated kinematics and type vectors. The event-level observables are incorporated as an additional input vector, $\mathcal{O}_{P_0} = E_P$, ensuring the encoder receives a complete description of the event $\mathcal{O}_P = \{\mathcal{O}_{P_0}, \mathcal{O}_{P_1}, \ldots, \mathcal{O}_{P_N}\}$. This input set is processed through a transformer encoder model that is position-equivariant, acknowledging that the particle objects within a collision event possess no intrinsic order. The encoder produces a set $X$ of $N$ elements, each being a $D_{\text{LATENT}}$ dimensional latent vector $x_i$, effectively mapping both the variable-length particle objects and fixed-length event observables into a unified latent space.

$$X = \{x_0, x_1, x_2, \ldots, x_N\} \text{ where } x_i = \text{TRANSFORMER}_{\text{ENCODER}}(\mathcal{O}_P)_i \in \mathbb{R}^{D_{\text{LATENT}}}. \quad (4)$$

Unlike in Ref. [18], the latent space of the VAE is coupled directly to the diffusion process. Instead of predicting an independent $\sigma$ from the VAE encoder, the learned diffusion noise schedule is used to determine the encoded vectors $x_0(0)$, and these noisy latent vectors are used for the decoder side of the VAE. The details and notation used to describe the diffusion process are presented below.

The decoder mirrors the encoder, employing a transformer to pre-process the encoded objects before outputting estimates for both the continuous kinematic features and the discrete object-type labels. Deep feed-forward multi-layer perceptrons (MLPs) are applied per-object to reconstruct these inputs, while the event observables are reconstructed using a separately parameterized MLP. The object-type MLP uses a softmax activation to produce a distribution over predicted object types.

$$
\begin{aligned}
w_i &= \text{TRANSFORMER}_{\text{DECODER}}([x_0(0), x_1(0), x_2(0), \ldots, x_N(0)])_i && \in \mathbb{R}^{D_{\text{LATENT}}} \\
\hat{P}_{P_i} &= MLP_P(w_i) \text{ for } i \geq 1 && \in \mathbb{R}^{D_{\text{PARTICLE}}} \\
\hat{T}_{P_i} &= \text{Softmax}(MLP_T(w_i)) \text{ for } i \geq 1 && \in \mathbb{R}^{D_{\text{TYPE}}} \\
\hat{E}_P &= MLP_E(w_0) && \in \mathbb{R}^{D_{\text{EVENT}}} \quad (5)
\end{aligned}
$$

## 3.2 Detector Encoder

The detector encoder employs an identical architecture to the Particle VAE encoder to encode the detector observations into the conditional latent space $Y$. The inputs are a cardinality M set of detector-level objects, each described by a vector of observables $P_{D_i}$ together with a one-hot encoding of the object type $T_{D_i}$. These features are concatenated $\mathcal{O}_{D_i} = (P_{D_i} || T_{D_i})$, and passed through the detector encoder. The output of this network is a set $Y$ of cardinality $M$ containing $D_{\text{LATENT}}$ dimensional latent detector vectors $y_i$

$$
Y = \{y_1, y_2, \ldots, y_M\} \text{ where } y_i = \text{TRANSFORMER}_{\text{DETECTOR}}([\mathcal{O}_{D_1}, \mathcal{O}_{D_2}, \ldots, \mathcal{O}_{D_M}])_i. \quad (6)
$$

## 3.3 Multiplicity Predictor

To accommodate the generation of variable-length particle-level events, a regression network that predicts the distribution of particle multiplicity conditioned on the encoded detector observations is used. This step is critical for determining the appropriate number of objects to generate, as the generative problem is not guaranteed to contain a one-to-one mapping between detector and truth level objects.

Starting with the encoded detector features $Y = \{y_1, y_2, \ldots, y_M\}$, a new learnable feature $y_0 \in D_{\text{LATENT}}$ is appended, and the set is processed with a transformer to extract multiplicity features $z$. Since the predictions are positive count values, a deep MLP is used to estimate the shape ($k$) and scale ($\theta$) parameters of a gamma distribution which can then be sampled from while unfolding an event.

$$
\begin{aligned}
z &= \text{TRANSFORMER}_{\text{MULTIPLICITY}}(\{y_0, y_1, y_2, \ldots, y_M\})_0 \\
\hat{N} &\sim \text{Gamma}(MLP_k(z), MLP_\theta(z)) \quad (7)
\end{aligned}
$$

## 3.4 Latent Diffusion Process

The conditional distribution $P(X|Y)$ is learned via a diffusion model, following the formulation presented in Ref. [18], based on the variational diffusion model interpretation first introduced in Ref. [50], extended to operate on sets of objects. The intrinsically unordered nature of sets

introduces a degree of ambiguity into the definition of the variable-length diffusion model and the training objective of the noise prediction model. These ambiguities are mitigated by imposing an arbitrary ordering on $X$ at the diffusion stage of the network via a total ordering function $O(x) \in \mathbb{R}$. In the following diffusion definitions, it will be assumed that $X$ is an ordered list of objects following the ordering function $O$, $X = [x_0, x_1, x_2, \ldots, x_N]$, such that for all $i < j$, $O(x_i) < O(x_j)$. This order will only affect the diffusion network, as all other components of the network are order equivariant.

The continuous time ($t \in [0,1]$) diffusion flow is extended to an element-wise generalization of traditional diffusion by defining the list distribution flow as

$$X(t) = [x_0(t), x_1(t), x_2(t), \ldots, x_n(t)] \text{ where } x_i(t) \sim \mathcal{N}(\alpha_i(t)x_i, \sigma_i(t)\mathbb{I}). \tag{8}$$

The traditional noise schedule of diffusion models is extended to multiple objects, with each position in the set defining its own schedule. The correct schedule is applied to the correct object by defining these schedules as a function of the object's position in $X$, which is fixed by the imposed ordering. A monotonic log signal-to-noise ratio (SNR) function, $\gamma_\phi(i, t)$, is learned, where $i \in \mathbb{N}$ and $t \in [0,1]$. As in Ref. [18], the function is parameterized as a positive definite neural network trained to minimize the variance of the diffusion loss term. The flow parameters based on these noise schedules are defined as

$$\sigma_i(t) = \sqrt{\text{sigmoid}(-\gamma_\phi(i, t))} \text{ and } \alpha_i(t) = \sqrt{\text{sigmoid}(\gamma_\phi(i, t))}.$$

As this formulation is an element-wise extension of the traditional diffusion process, the forward and backward dynamics remain identical to the original VLD, although applied element-wise with each position's noise schedule. A system of Stochastic Differential Equations (SDEs) is defined as

$$dx_i(t) = f(x_i, t)dt + g(t)d\mathbf{w} \qquad \text{(Forward SDE)}$$

$$dx_i(t) = \left[ f(x_i, t) + g^2(t) \frac{\hat{\epsilon}_\theta(x_i(t), X(t), Y, t)}{\sigma_i(t)} \right] dt + d\bar{\mathbf{w}} \qquad \text{(Reverse SDE)}$$

for each $i \in \{0, 1, 2, \ldots, N\}$. A noise-prediction formulation is employed for the score network, with $\nabla_{x_i} \log p(x_i) = -\frac{1}{2}\hat{\epsilon}_\theta(x_i(t), X(t), Y, t)$ [45]. A key detail is that the denoising network (detailed below) depends not only on the conditioning $Y$ and the current noisy latent vector $x_i(t)$, but on all other noisy latent samples in $X(t)$ as well. This allows different generative components to share information with each other. This contextual information allows the denoising network to adjust its predictions based on the other objects currently being generated. Without these inputs, the denoising network could confuse objects with each other when the signal-to-noise ratio is low.

### 3.4.1 Particle Denoising Network

To address the challenges of modeling variable-length data and effectively predicting the noise in the context of diffusion models, a noise prediction function employing transformers, $\hat{\epsilon}_\theta(x_i, X, Y, t)$, is used to learn a novel list-generative and set-conditioned denoising network. This network leverages an encoder-only transformer architecture, processing all noisy latent vectors $x_i(t)$ in parallel. By employing attention mechanisms, the network integrates both the particle-level inputs, $X$, and the conditioning detector-level inputs, $Y$, enriching the denoising process with comprehensive contextual information.

For particle-level inputs, $X(t) = [x_0(t), x_1(t), \ldots, x_N(t)]$, Fourier positional features $P_i$ are incorporated, following the method introduced in Ref. [51]. Additionally, a unique learned

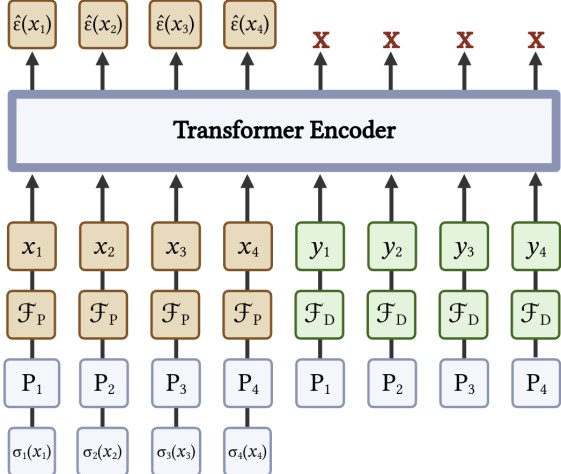

Figure 2: A simplified block diagram of the denoising network and all of the inputs.

*flag vector*, $\mathcal{F}_{\mathcal{P}} \in \mathbb{R}^N$, is appended to identify these inputs as noisy particle-level data. The network may need to adjust its outputs based on the specific stage of the diffusion process, so the current noise scale, $\sigma_i(t)$, is additionally included in the inputs.

A parallel pre-processing pipeline is applied to the conditioning set $Y$, adding position information and the detector-level flag vector to this input, $\mathcal{F}_{\mathcal{D}} \in \mathbb{R}^M$. Both sets of pre-processed inputs are then transformed via MLPs which map these inputs into the denoising transformer's hidden dimensionality, $D_{\text{DENOISE}}$, This produces two lists of $D_{\text{DENOISE}}$ dimensional vectors, one of length $N$ representing the particle level event at a time step $t$, and the other of length $M$ representing the detector level event,

$$\mathcal{P} = [\mathcal{P}_0, \mathcal{P}_1, \mathcal{P}_2, \ldots, \mathcal{P}_N] \qquad \text{where } \mathcal{P}_i = MLP_{\mathcal{P}}(x_i || \mathcal{F}_{\mathcal{P}} || P_i || \sigma_i(t)),$$
$$\mathcal{D} = [\mathcal{D}_1, \mathcal{D}_2, \ldots, \mathcal{D}_M] \qquad \text{where } \mathcal{D}_i = MLP_{\mathcal{D}}(y_i || \mathcal{F}_{\mathcal{D}} || P_i).$$

The denoising network is then defined as the output of a transformer encoder with the two lists used as inputs. A block diagram of the denoising network and its inputs is shown in Figure 2. The noise predictions are extracted by dropping the transformer outputs for the detector-level inputs and indexing the outputs by particle position.

$$\hat{\epsilon}_\theta(x_i, X, Y, t) = \text{TRANSFORMER}(\mathcal{P} || \mathcal{D})_i \tag{9}$$

### 3.4.2 Noise Schedule Network

The noise schedule $\log SNR$ is parameterized as a monotonically decreasing function with respect to time, conditioned on position, $\gamma_\phi(i, t)$. Following [50], this monotonic function is implemented via a neural network with positive weights and monotonic activations. Weights are forced positive by squaring them before computing the linear operation, and sigmoid activations are used for the hidden layers. Inputs are the time as a scalar, $t \in [0, 1]$, and the position encoded as learned $D_{\text{SCHEDULE}}$ dimensional position vectors $P_i \in \mathbb{R}^{D_{\text{SCHEDULE}}}$. The network's hidden layer contains $1{,}000$ dimensions to allow for nonlinear behaviour, and the final layer outputs a scalar value for the $\log SNR$.

Independently, the end-points of the noise schedule $\gamma_{min}$ and $\gamma_{max}$, are learned and held identical across all positions. The network learns an unconstrained schedule which is then rescaled to fit into the $\gamma$ bounds. Given $W_1 \in \mathbb{R}^{1000 \times (D_{\text{SCHEDULE}}+1)}$, $W_2 \in \mathbb{R}^{1 \times 1000}$, and $\sigma(\_)$ the sigmoid function, the schedule is defined as:

$$\tilde{\gamma}_{\phi}(t,i) = -W_2^2 \sigma\left(W_1^2 [t||P_i]\right) \tag{10}$$

$$\gamma_{\phi}(t,i) = (\gamma_{max} - \gamma_{min})\frac{\tilde{\gamma}_{\phi}(t,i) - \tilde{\gamma}_{\phi}(0,i)}{\tilde{\gamma}_{\phi}(1,i) - \tilde{\gamma}_{\phi}(0,i)} + \gamma_{min}. \tag{11}$$

## 3.5 Training

The diffusion model is trained with an end-to-end variational inference approach following [18]. All networks are trained simultaneously via a unified loss function known as the evidence lower bound (ELBO). Particle-level unfolding introduces several additional aspects to this diffusion process, notably variable-length outputs and the need for a multiplicity predictor. The reconstruction and prior terms remain from the traditional VAE ELBO [49], and the denoising loss is reinterpreted as the diffusion prior as in Ref. [50]. A final generative distribution is added to account for the multiplicity output. The full generalized ELBO is:

$$
\begin{aligned}
\mathcal{L} = &\sum_{i \in \{0,1,...N\}} D_{KL}[q(x_i(1)|\mathcal{O}_P, \mathcal{O}_D) \,\|\, p(x_i(1))] && \text{\small PRIOR LOSS} \\
&+ \sum_{i \in \{0,1,...N\}} \mathbb{E}_{q(x_i(0)|\mathcal{O}_P)}\left[-\log p(\hat{\mathcal{O}}_P|x_i(0)\right] && \text{\small RECONSTRUCTION LOSS} \\
&+ \sum_{i \in \{0,1,...N\}} \mathbb{E}_{\epsilon \sim \mathcal{N}(\mathbf{0},\mathbb{I}), t \sim \mathcal{U}(0,1)}\left[\gamma_{\phi}'(t)\,\|\epsilon - \hat{\epsilon}_{\theta}(x_i(t),X(t),Y,t)\|_2^2\right] && \text{\small DENOISING LOSS} \\
&- \log p(\hat{N} = N|\mathcal{O}_D). && \text{\small MULTIPLICITY LOSS} \quad (12)
\end{aligned}
$$

PRIOR LOSS:    The prior loss is an element-wise extension of the regular VDM prior loss [50], matching each final-time element to the prior distribution independently. A standard normal prior, $p(x_i(1)) \sim \mathcal{N}(0,1)$, is used for all latent vectors.

RECONSTRUCTION LOSS:    The reconstruction loss is extended to an element-wise version of a regular VAE reconstruction. This is complicated by the fact that there are several different outputs for every element, and a special event-level element which may also have several outputs. A Gaussian likelihood is assumed for the continuous outputs and a multinomial likelihood for the type predictions. Assuming a Gaussian likelihood for the continuous outputs and a multinominal likelihood for the type predictions, this sector of the loss function expands as

$$
\begin{aligned}
\sum_{i \in \{0,1,...N\}} \mathbb{E}_{q(x_i(0)|\mathcal{O}_P)}\left[-\log p(\hat{\mathcal{O}}_P|x_i(0))\right] = &\left\|\hat{E}_P - E_P\right\|_2^2 \\
&+ \sum_{i \in \{1,2,...N\}} \left\|\hat{P}_{P_i} - P_{P_i}\right\|_2^2 \\
&+ \sum_{i \in \{1,2,...N\}} \sum_{D_{\text{TYPE}}} T_{P_i} \log \hat{T}_{P_i}. \tag{13}
\end{aligned}
$$

DENOISING LOSS:    The denoising loss extends element-wise to the multi-object case. As the signal-to-noise ratio reduces, the noisy inputs $x_i(t)$ may become ambiguous with respect to each other, ultimately culminating in all noisy inputs looking identical under the prior distribution at $t = 1$. Therefore, in order to use the mean squared error loss function in a well defined manner for high values of $t$, it is required that the inputs remain ordered and labeled, breaking the position equivariance of the VAE encoder and decoder. This symmetry breaking occurs only at

the diffusion step, and this identification additionally allows independent noise schedules to be learned for each position.

MULTIPLICITY LOSS: The multiplicity estimate $\hat{N}$ follows a gamma distribution (Eq. 7), and employs a gamma likelihood when computing the corresponding loss term. Using $\hat{k} = MLP_k(z)$ and $\hat{\theta} = MLP_\theta(z)$ ,

$$-\log p(\hat{N} = N|\mathcal{O}_D) = -\hat{k}\log\hat{\theta} + \log\Gamma(\hat{k}) - \hat{k}\log N + N\hat{\theta} + \log N. \qquad (14)$$

## 3.6 Inference

The generative process must be slightly adapted during inference to account for the variable number of outputs. Given only a set of detector observables $\mathcal{O}_D$, the generative inference process may be described as follows:

1. Encode detector observable, $Y = \text{TRANSFORMER}_{\text{DETECTOR}}(\mathcal{O}_D)$.

2. Extract multiplicity latent vector, $z = \text{TRANSFORMER}_{\text{MULTIPLICITY}}(y_0||Y)_0$.

3. Sample a multiplicity, $\hat{N} \sim \Gamma(MLP_k(z), MLP_\theta(z))$, and round to the nearest integer $N = [\hat{N}]$.

4. Sample $N$ standard normal vectors from the prior distribution, for $i \in \{0, 1, \ldots, N\}$, $x_i(1) \sim \mathbb{N}(0, 1)$.

5. Perform reverse diffusion process using an ODE solver [52] to predict the denoised latent $x_i(0)$ following the independently learned noise schedule $\gamma_\phi(i, t)$ for each element.

6. Predict the final particle-level observables by decoding the denoised latents following Eq 5.

# 4 Example Use-case: Semi-leptonic $t\bar{t}$ Unfolding

As an example of a common use-case, VLD is used to unfold proton-proton collision events containing top quark pairs ($t\bar{t}$). The semi-leptonic decay mode is chosen, in which one top quark decays to three quarks via $t \to Wb \to qqb$ and the other to a lepton, neutrino and $b$-quark via $t \to Wb \to \ell\nu b$. A Feynman diagram that contributes to the process is shown in Figure 3. This decay mode, with a final state containing one lepton, two $b$-quarks, two light quarks, and missing momentum originating from the neutrino (which escapes the detector without interaction), is an excellent test case due to its complexity and importance for precision measurements and searches for new physics [53–63].

## 4.1 Dataset

Simulated $t\bar{t}$ events from proton-proton collisions are generated with the Standard Model (SM) at a centre-of-mass energy of $\sqrt{s} = 13$ TeV using MADGRAPH_AMC@NLO [64] (v3.4.2, NCSA license) for the matrix element calculation and PYTHIA8 [65] (v8.306, GPL-2) for the parton showering and hadronization. Interaction with the experimental apparatus is simulated with DELPHES [66] (v3.5.0, GPL-3) using the default CMS detector card.

   Electrons and muons are required to have a transverse momentum $p_T > 25$ GeV and absolute pseudo-rapidity $|\eta| < 2.5$. The light and $b$-quarks are reconstructed as *jets*, collimated energy deposits grouped using the anti-$k_T$ [67] algorithm with a radius parameter of $R = 0.5$,

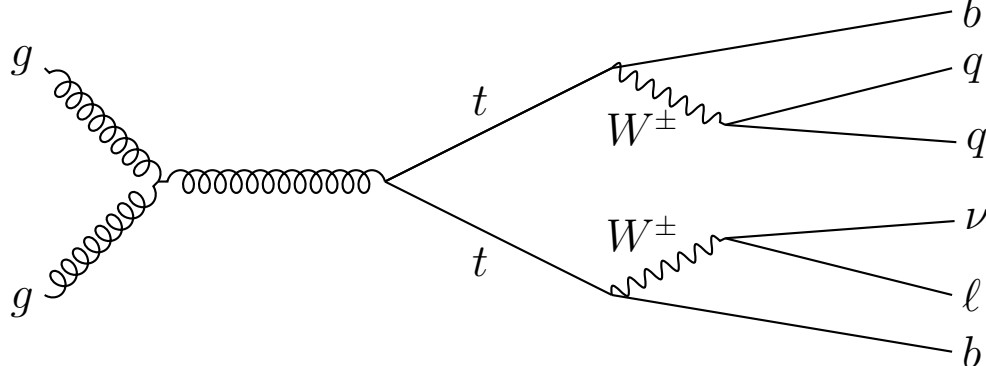

Figure 3: A representative Feynman diagram of $t\bar{t}$ production in the semi-leptonic decay mode.

which must satisfy the same $p_T$ and $\eta$ requirements as the leptons. Jets originating from $b$-quarks are tagged as such with a $p_T$-dependent efficiency. At particle level, the jet algorithm is applied to stable particles rather than energy deposits, and jets containing $b$-quarks are directly tagged as such.

Selected events are required to have one electron or muon and at least four jets, of which at least two are $b$-tagged, at both particle and detector level. 14M and 1M events are used for training and testing, respectively. Potential adjustments needed to account for events which pass one, but not both, of these selections are discussed in Section 5.

An additional sample of approximately 2M events is produced to explore the impact of the training sample prior. This sample uses the dim6top_LO_UFO [68] model to incorporate new physics in the top-gluon vertex via the $c_{tg}$ parameter, which is set to a value of 25. All other settings and the event selection are identical to the SM sample.

Particle-level observables are represented by the vector $P_i = (P_x, P_y, P_z, \log(E+1), \log(M+1))$. Both the mass and energy are included in the representation to improve robustness to numerical issues. Particles are categorized into four types: light-quark jets, $b$-quark jets, electrons, and muons, represented as a four dimensional one-hot type vector, $T_{P_i} \in \{0, 1\}^4$. The event-level observables for a particle-level event are taken to be the magnitude of the missing transverse momentum $E_T^{\text{miss}}$ and its azimuthal angle $\phi^{\text{miss}}$, along with the neutrino kinematics $(P_x^\nu, P_y^\nu, P_z^\nu, E^\nu)$. At detector level, two coordinate-representations of the four-momentum are provided; $P_{D_i}^{\text{Cart}} = (P_x, P_y, P_z, \log(E+1))$ and $P_{D_i}^{\text{Polar}} = (P_T, \eta, \sin\phi, \cos\phi, \log(M+1))$, as well as the type one-hot vector, $T_{D_i}$, for each object. The event-level observables are only $E_T^{\text{miss}}$ and $\phi^{\text{miss}}$, as the neutrino kinematics are unobserved at detector level.

## 4.2 Standard Model Results

The VLD's learned log signal-to-noise ratio function $\gamma_\phi(i, t)$ and corresponding $\beta$ schedule after training on the SM $t\bar{t}$ dataset are shown in Figure 4. Low (high) values of the signal-to-noise ratio indicate high (low) amounts of added noise. The objects are ordered by decreasing $p_T$, while the "Event" object corresponds to the event-level observables. Early in the generation process, at high but decreasing values of $t$, the learned signal-to-noise ratio increases most quickly for the softest objects. This can be understood as the model adjusting the amount of added noise to be correctly proportional to the size of the energy and momentum kinematic quantities for each object. During the bulk of the generation process at intermediate values of $t$, the model increases the SNR at roughly equal rates for all objects in the event.

The VLD model performance is evaluated on a testing sample, generated identically to the

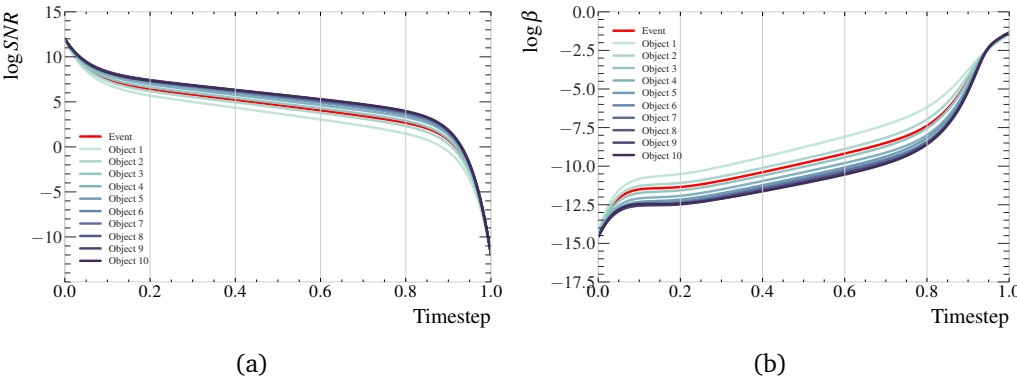

Figure 4: Example learned noise schedule for each object. Independent noise schedules are learned for each unfolded object, ordered by true $p_T$. The "Event" represents the event level observables. Shown are (a) the signal-to-noise ratio (SNR) learned during training and (b) the equivalent $\beta$ schedule used during inference, following the DDPM framework [45].

training sample. Figures 5 and 6 show the kinematic distributions of the leptons and jets respectively[2]. The distributions are inclusive over all leptons or jets in the events. Distributions labelled "truth" are from events without detector effects, the target of the unfolding. Distributions labeled "detector" are from events which include simulated effects of the detector and are used to condition generation. The distributions labelled "unfolded" are from particle level events generated by the VLD unfolding algorithm.

In general, there is excellent agreement between the unfolded and truth distributions for the inclusive lepton and jet kinematics. This is expected, as the network is trained to directly predict these quantities. Some disagreement is found at the kinematic edges, for example at low $p_T$ and mass or at extreme values of $\eta$, due to a lack of examples of events migrating across the selection requirements from particle to detector level. This could be mitigated by imposing a tighter selection in generation than in training, avoiding the need to learn sharp cut-offs in the target distributions. Table 1 displays three measures of distance[3] between the truth distributions and the corresponding detector and unfolded distributions for the jet and lepton kinematics. In most cases, the distance to the truth distribution is smaller for the unfolded distributions than the detector distributions. One exception is in the jet $\eta$ observable, due to events migrating across the event selection between particle and detector levels at high $|\eta|$.

Distributions of the event-level observables $E_T^{\mathrm{miss}}$, $\phi^{\mathrm{miss}}$, and the neutrino pseudo-rapidity $\eta_\nu$ are shown in Fig. 7. These show good closure except for a slight peak near zero in $\eta_\nu$. Since $\eta_\nu$ is not measurable at detector level, conditioning on this observable for the diffusion process is relatively weak. It is then unsurprising that the network tends to return the mean value of $\eta_\nu$ in events that are particularly difficult to unfold.

A crucial requirement for full-event unfolding is the capacity to accommodate variable object multiplicities, such as the number of reconstructed jets. Four jets are expected from the quarks produced from the $t\bar{t}$ decay (see Fig. 3), but some jets may fail the selection requirements and additional jets can be generated from other activity such as initial- or final-state radiation. Figure 8 shows the distribution of the jet multiplicity for the truth-level, detector-level, and unfolded events, along with $H_T$, the scalar sum of all $p_T$ in the event. There is excellent agreement in the jet multiplicity distribution, indicating that the network is correctly accounting for the variable dimensionality of the unfolding task. However, a slight disagreement between

---

[2]Excepting the lepton masses, whose values are well known and need not be unfolded.

[3]Measure definitions are in Appendix C of Ref. [18].

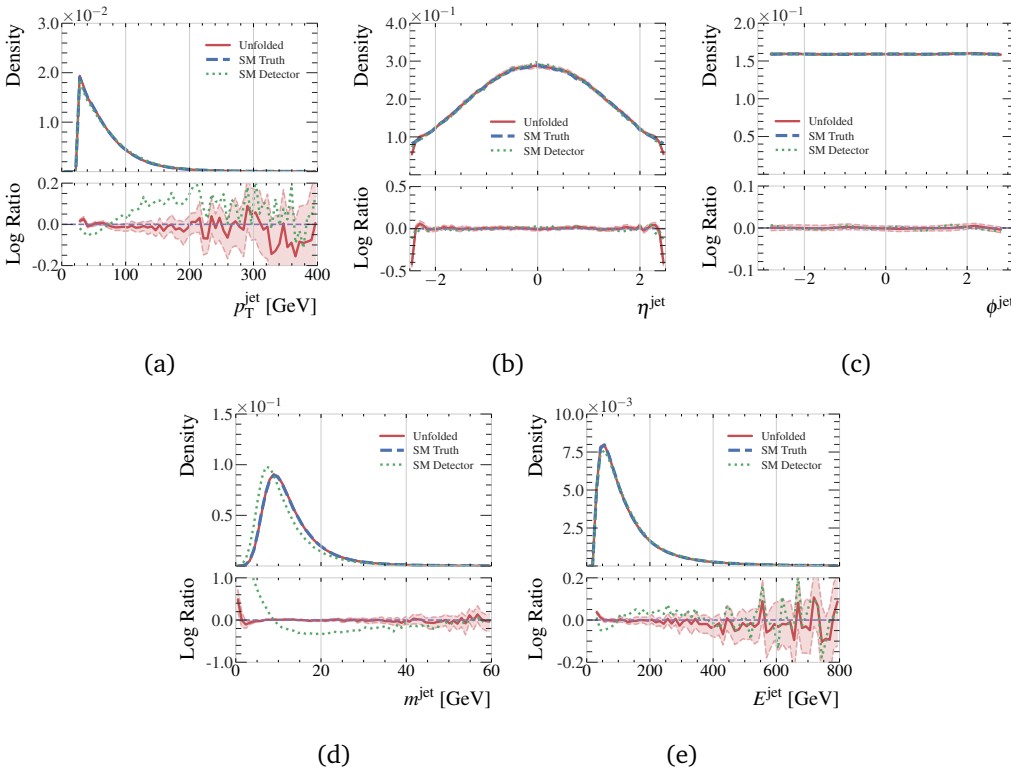

Figure 5: Inclusive kinematic distributions for jets in the SM testing dataset, comparing the true particle-level jets (dashed blue), the unfolded particle-level jets (solid red), and the detector-level jets (dotted green). The unfolded distributions include error bounds estimated by sampling each event 128 times.

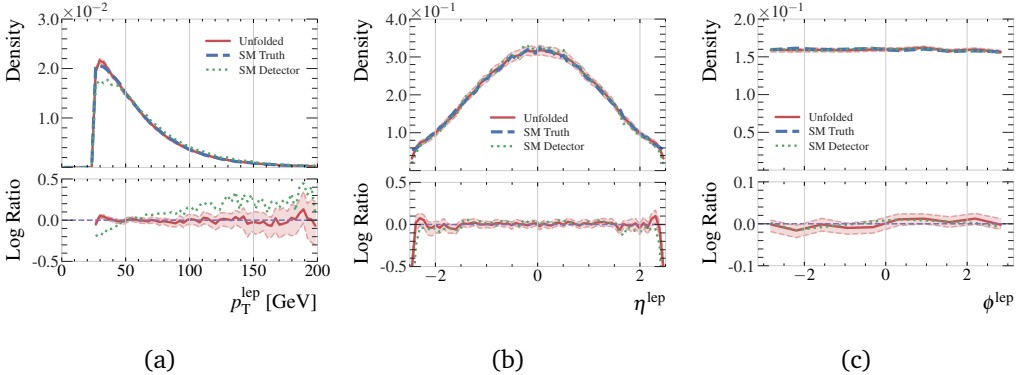

Figure 6: Inclusive kinematic distributions of leptons in the SM testing dataset, comparing the true particle-level leptons (dashed blue), the unfolded particle-level leptons (red solid), and the detector-level leptons (dotted green). Unfolded distributions include error bounds estimated by sampling each event 128 times.

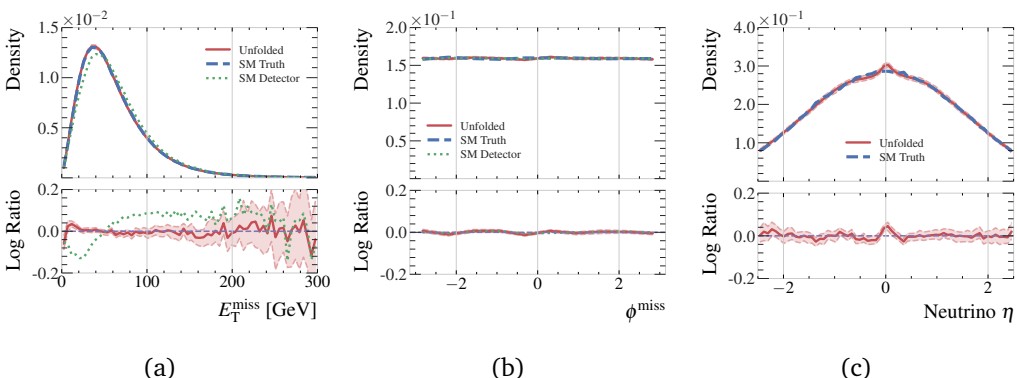

Figure 7: Event-level quantities in the SM testing dataset, comparing the true particle-level (dashed blue), the unfolded particle-level (solid red), and the detector-level (dotted green). Unfolded distributions include error bounds estimated by sampling each event 128 times.

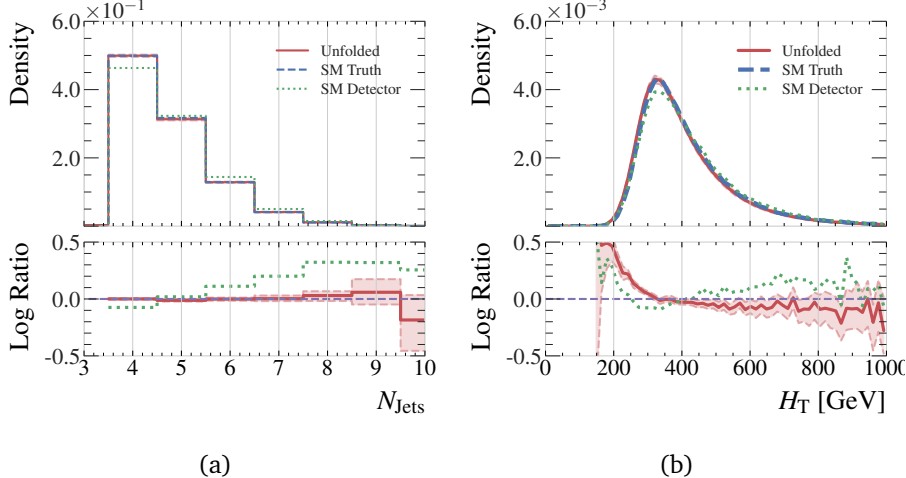

Figure 8: Distributions of (a) jet multiplicity and (b) $H_T$ comparing the true particle-level events (dashed blue), the unfolded particle-level events (red solid), and the detector-level events (dotted green). Unfolded distributions include error bounds estimated by sampling each event 128 times.

Table 1: Wasserstein, Energy, and Kulback-Leibler distance measures between truth and detector or truth and unfolded top quark kinematic distributions. The unfolded distribution uncertainties are estimated by sampling each event 128 times.

| Observable | Level | Wasserstein | Energy | KL |
|---|---|---|---|---|
| $p_T^{\mathrm{Jet}}$ | Unfolded | $0.36 \pm 0.08$ | $0.04 \pm 0.01$ | $0.02 \pm 0.01$ |
| | Detector | $2.30$ | $0.27$ | $0.26$ |
| $\eta^{\mathrm{Jet}}$ | Unfolded | $0.01 \pm 0.00$ | $0.00 \pm 0.00$ | $13.03 \pm 1.18$ |
| | Detector | $0.01$ | $0.01$ | $3.44$ |
| $\phi^{\mathrm{Jet}}$ | Unfolded | $0.00 \pm 0.00$ | $0.00 \pm 0.00$ | $0.01 \pm 0.01$ |
| | Detector | $0.00$ | $0.00$ | $0.02$ |
| $m^{\mathrm{Jet}}$ | Unfolded | $0.03 \pm 0.01$ | $0.01 \pm 0.00$ | $0.17 \pm 0.05$ |
| | Detector | $1.52$ | $0.52$ | $61.66$ |
| $E^{\mathrm{Jet}}$ | Unfolded | $0.66 \pm 0.26$ | $0.05 \pm 0.02$ | $0.01 \pm 0.00$ |
| | Detector | $2.05$ | $0.20$ | $0.06$ |
| $p_T^{\mathrm{Lepton}}$ | Unfolded | $0.27 \pm 0.10$ | $0.05 \pm 0.02$ | $0.17 \pm 0.05$ |
| | Detector | $3.89$ | $0.53$ | $2.64$ |
| $\eta^{\mathrm{Lepton}}$ | Unfolded | $0.01 \pm 0.00$ | $0.01 \pm 0.00$ | $16.72 \pm 3.27$ |
| | Detector | $0.03$ | $0.02$ | $56.01$ |
| $\phi^{\mathrm{Lepton}}$ | Unfolded | $0.01 \pm 0.01$ | $0.01 \pm 0.00$ | $0.08 \pm 0.05$ |
| | Detector | $0.01$ | $0.01$ | $0.02$ |
| $E_T^{\mathrm{miss}}$ | Unfolded | $0.31 \pm 0.06$ | $0.04 \pm 0.01$ | $0.04 \pm 0.01$ |
| | Detector | $3.03$ | $0.41$ | $0.95$ |
| $H_T$ | Unfolded | $7.45 \pm 0.54$ | $0.51 \pm 0.03$ | $0.20 \pm 0.02$ |
| | Detector | $8.54$ | $0.59$ | $0.20$ |

the unfolded and truth distributions is seen at low $H_T$. To further interrogate the robustness of the jet multiplicity prediction, Fig. 9 shows the $H_T$, jet $p_T$, and jet mass distributions binned in jet multiplicity, with consistent performance shown in all bins including the mis-modeling at low values of $H_T$. The network's ability to reproduce the truth-level kinematic distributions is therefore found to be independent of the jet multiplicity. The mis-modeling at low $H_T$ may be a consequence of the limited number of training examples in this region of phase space. Possible solutions to this issue are discussed in Section 5.

The defining feature of full-event unfolding is the capacity to obtain an unfolded distribution of an arbitrary new observable, such as those defined on reconstructed objects like top quarks, which are functions of the lower-level jet and lepton observables. The pseudo-top algorithm [69,70] is used to reconstruct hadronically- and leptonically-decaying top quark candidates, and check the performance of our unfolding on the kinematics of these reconstructed systems. Unlike for classical unfolding algorithms, a change in the reconstruction algorithm does not obsolete the results, as these can be simply recalculated with the new algorithm.

Figure 10 shows the kinematic distributions for the reconstructed hadronically-decaying top quark, leptonically-decaying top quark, and the $t\bar{t}$ system. Closure in these distributions is not as good as for the kinematics of the jets and leptons, though it is not surprising that these observables are more difficult to model as they were not directly used to optimize the networks. In particular, the top quark mass distributions are very difficult to unfold, with the sharp peaks at truth level not reproduced after unfolding. This is an illustration of the well known difficulty in modeling sharp features with generative models [12].

Table 2 shows the distance metrics computed for the kinematics of the hadronically-decaying top quark, the leptonically-decaying top quark, and the $t\bar{t}$ system. In some cases, the unfolded

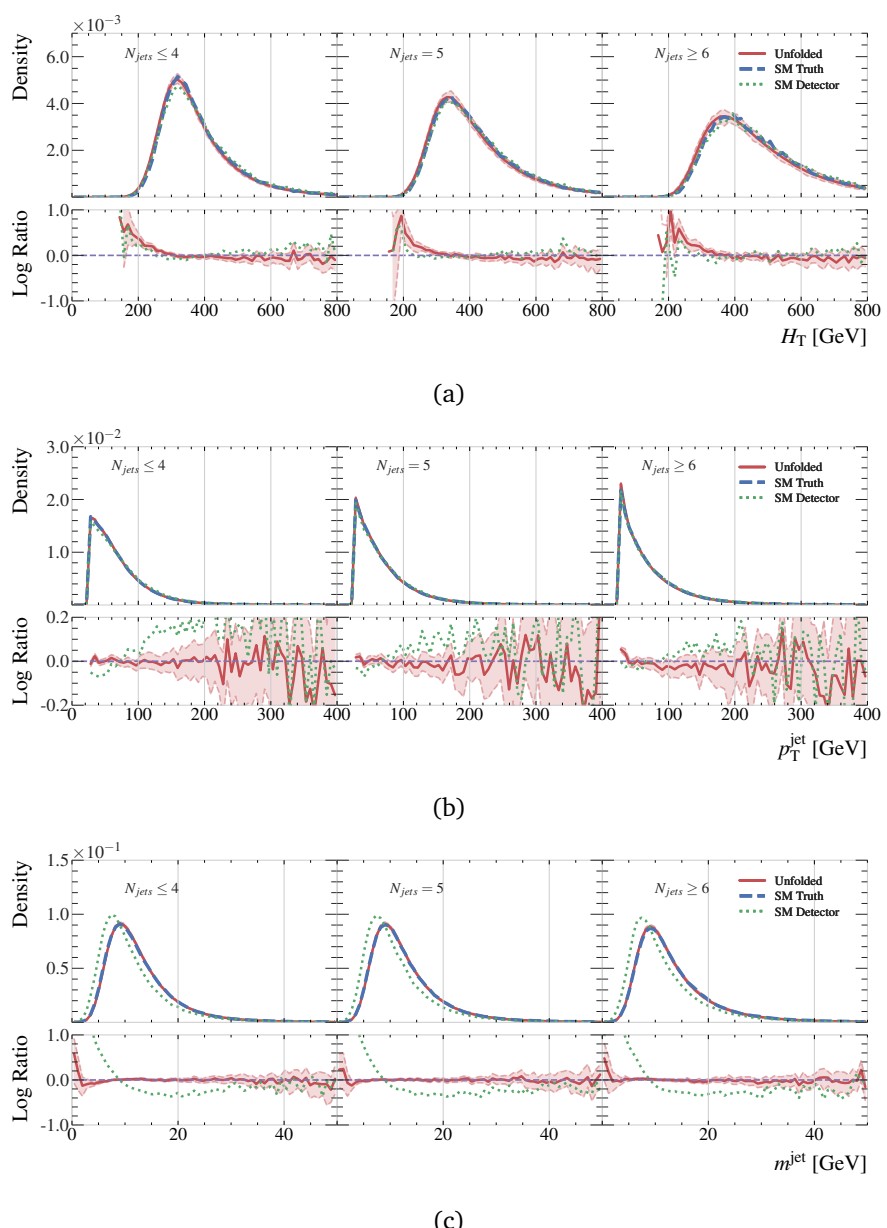

Figure 9: Distributions of (a) $H_{\mathrm{T}}$, (b) the inclusive jet $p_{\mathrm{T}}$ and (c) jet mass, each in bins of jet multiplicity in the SM testing dataset. Shown are true particle-level jets (dashed blue), the unfolded particle-level jets (solid red), and the detector-level jets (dotted green). Unfolded distributions include error bounds estimated by sampling each event 128 times.

distributions are slightly further from the truth distributions than the detector-level distributions, suggesting the unfolding is not correctly modeling the subtle correlations between the objects in particle-level events. For the $p_{\mathrm{T}}$ of the hadronically-decaying top quark and the $t\bar{t}$ system, the distance between the unfolded and truth distributions is larger than the distance between the detector-level and truth distributions, while for the corresponding energy distributions, it is smaller. Meanwhile, the opposite is observed for the leptonically-decaying top quark. These distributions are highly correlated with the top quark mass, so a promising avenue for future work would be to incorporate physics constraints, such as knowledge of the expected top mass value, in order to improve these observables. In Ref. [18], in which the parton-level top quark kinematics were directly included in the training objective, the modeling of the top quark mass peaks is improved through use of a physics-inspired consistency loss. Unfortunately this strategy is not immediately applicable in particle-level unfolding, where the network does not directly predict the top kinematics. Such considerations are discussed further in Section 5.

To further probe the performance of VLD based unfolding, additional event-level observables of interest in $t\bar{t}$ production [59] are constructed, shown in Figure 11[4]. There is remarkably good agreement between truth and unfolded in most of these complex observables, with agreement within uncertainties almost everywhere.

To probe the source of the non-closure in the top quark kinematics, the top quark[5] $p_{\mathrm{T}}$ and $t\bar{t}$ mass distributions are reconstructed in bins of jet multiplicity in Figure 12, along with the top quark $p_{\mathrm{T}}$ in bins of the $t\bar{t}$ mass. The disagreement of the top quark $p_{\mathrm{T}}$ seen in Figure 10 is reproduced in all jet multiplicity bins, indicating that this disagreement is not being produced by the variable-length nature of the unfolding. In contrast, the disagreement of the top quark $p_{\mathrm{T}}$ gets worse as a function of increasing $t\bar{t}$ mass. High-mass events are rare in the training set (see Fig. 10i), so this non-closure may be due to limited training examples, as with the non-closure at low $H_{\mathrm{T}}$. Effects due to the choice of prior distributions could be overcome via specific choices in training dataset construction, which need not match the observed data. This is an important feature, not only for coverage of extreme regions of phase space, but also to ensure a model trained on SM events does not simply reproduce SM truth distributions and wash away any new physics that might be present in data. The latter point is investigated in the following section.

---

[4]Full definitions of these variables are given in Appendix A.

[5]In these $p_{\mathrm{T}}$ distributions, both the leptonically- and hadronically-decaying top quarks are included.

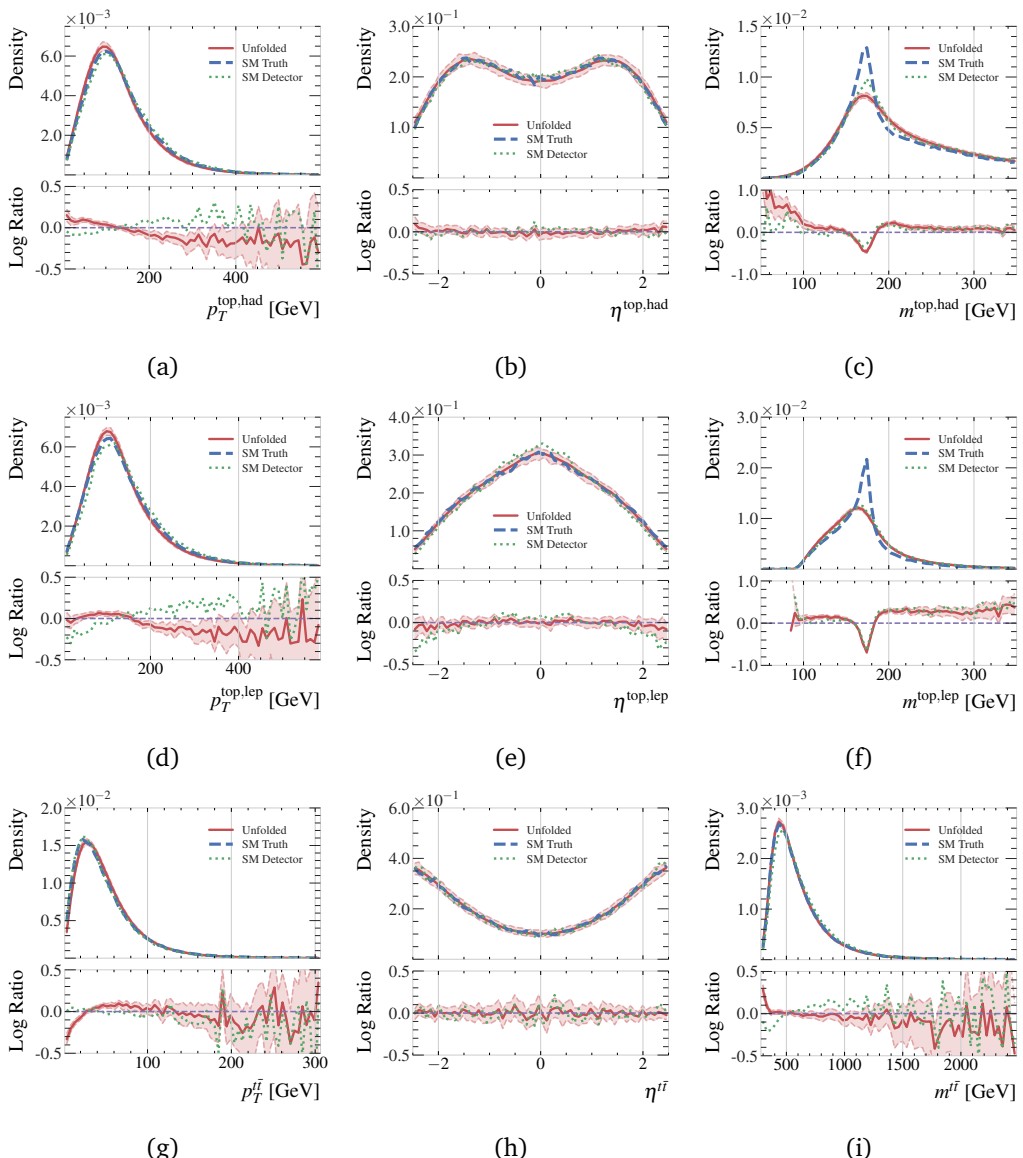

Figure 10: Distributions of reconstructed top quark and $t\bar{t}$ system kinematics in the SM testing dataset. Shown are true particle-level (dashed blue), the unfolded particle-level (solid red), and the detector-level (dotted green). Unfolded distributions include error bounds estimated by sampling each event 128 times.

Table 2: Wasserstein, Energy, and Kulback-Leibler distance measures between truth and detector or truth and unfolded top quark kinematic distributions. The unfolded distribution uncertainties are estimated by sampling each event 128 times.

| Observable | Level | Wasserstein | Energy | KL |
|---|---|---|---|---|
| $p_{\mathrm{T}}^{\mathrm{top,had}}$ | Unfolded | $5.71 \pm 0.40$ | $0.49 \pm 0.03$ | $0.34 \pm 0.05$ |
| | Detector | $4.25$ | $0.39$ | $0.25$ |
| $\eta^{\mathrm{top,had}}$ | Unfolded | $0.01 \pm 0.00$ | $0.01 \pm 0.00$ | $6.76 \pm 2.34$ |
| | Detector | $0.01$ | $0.01$ | $8.40$ |
| $\phi^{\mathrm{top,had}}$ | Unfolded | $0.01 \pm 0.01$ | $0.00 \pm 0.00$ | $3.74 \pm 1.35$ |
| | Detector | $0.01$ | $0.01$ | $5.44$ |
| $E^{\mathrm{top,had}}$ | Unfolded | $2.51 \pm 1.14$ | $0.10 \pm 0.05$ | $0.02 \pm 0.01$ |
| | Detector | $10.92$ | $0.60$ | $0.08$ |
| $m^{\mathrm{top,had}}$ | Unfolded | $4.63 \pm 0.32$ | $0.55 \pm 0.04$ | $4.46 \pm 0.25$ |
| | Detector | $4.05$ | $0.50$ | $2.37$ |
| $p_{\mathrm{Out}}^{\mathrm{top,had}}$ | Unfolded | $0.82 \pm 0.15$ | $0.14 \pm 0.02$ | $0.42 \pm 0.10$ |
| | Detector | $0.48$ | $0.06$ | $0.28$ |
| $p_{\mathrm{T}}^{\mathrm{top,lep}}$ | Unfolded | $4.07 \pm 0.30$ | $0.36 \pm 0.03$ | $0.28 \pm 0.04$ |
| | Detector | $8.12$ | $0.73$ | $0.82$ |
| $\eta^{\mathrm{top,lep}}$ | Unfolded | $0.01 \pm 0.00$ | $0.01 \pm 0.00$ | $5.79 \pm 2.00$ |
| | Detector | $0.05$ | $0.04$ | $51.77$ |
| $\phi^{\mathrm{top,lep}}$ | Unfolded | $0.01 \pm 0.01$ | $0.01 \pm 0.01$ | $3.36 \pm 1.17$ |
| | Detector | $0.01$ | $0.00$ | $4.54$ |
| $E^{\mathrm{top,lep}}$ | Unfolded | $7.41 \pm 0.88$ | $0.40 \pm 0.04$ | $0.08 \pm 0.02$ |
| | Detector | $4.88$ | $0.36$ | $0.12$ |
| $m^{\mathrm{top,lep}}$ | Unfolded | $4.50 \pm 0.19$ | $0.55 \pm 0.01$ | $7.99 \pm 0.31$ |
| | Detector | $4.56$ | $0.60$ | $7.80$ |
| $p_{\mathrm{Out}}^{\mathrm{top,lep}}$ | Unfolded | $1.02 \pm 0.15$ | $0.17 \pm 0.02$ | $0.57 \pm 0.12$ |
| | Detector | $0.27$ | $0.03$ | $0.26$ |
| $m^{t\bar{t}}$ | Unfolded | $7.58 \pm 1.58$ | $0.33 \pm 0.08$ | $0.04 \pm 0.01$ |
| | Detector | $16.11$ | $0.92$ | $0.15$ |
| $p_{\mathrm{T}}^{t\bar{t}}$ | Unfolded | $1.63 \pm 0.15$ | $0.26 \pm 0.02$ | $0.95 \pm 0.14$ |
| | Detector | $1.24$ | $0.14$ | $0.26$ |
| $\eta^{t\bar{t}}$ | Unfolded | $0.01 \pm 0.01$ | $0.01 \pm 0.01$ | $13.56 \pm 5.91$ |
| | Detector | $0.01$ | $0.01$ | $27.63$ |
| $\phi^{t\bar{t}}$ | Unfolded | $0.01 \pm 0.01$ | $0.01 \pm 0.01$ | $5.75 \pm 1.76$ |
| | Detector | $0.01$ | $0.00$ | $7.67$ |
| $\chi^{t\bar{t}}$ | Unfolded | $0.11 \pm 0.02$ | $0.05 \pm 0.01$ | $5.41 \pm 1.47$ |
| | Detector | $0.09$ | $0.04$ | $4.52$ |
| $y_{boost}^{t\bar{t}}$ | Unfolded | $0.01 \pm 0.00$ | $0.01 \pm 0.00$ | $22.50 \pm 6.51$ |
| | Detector | $0.02$ | $0.03$ | $80.18$ |

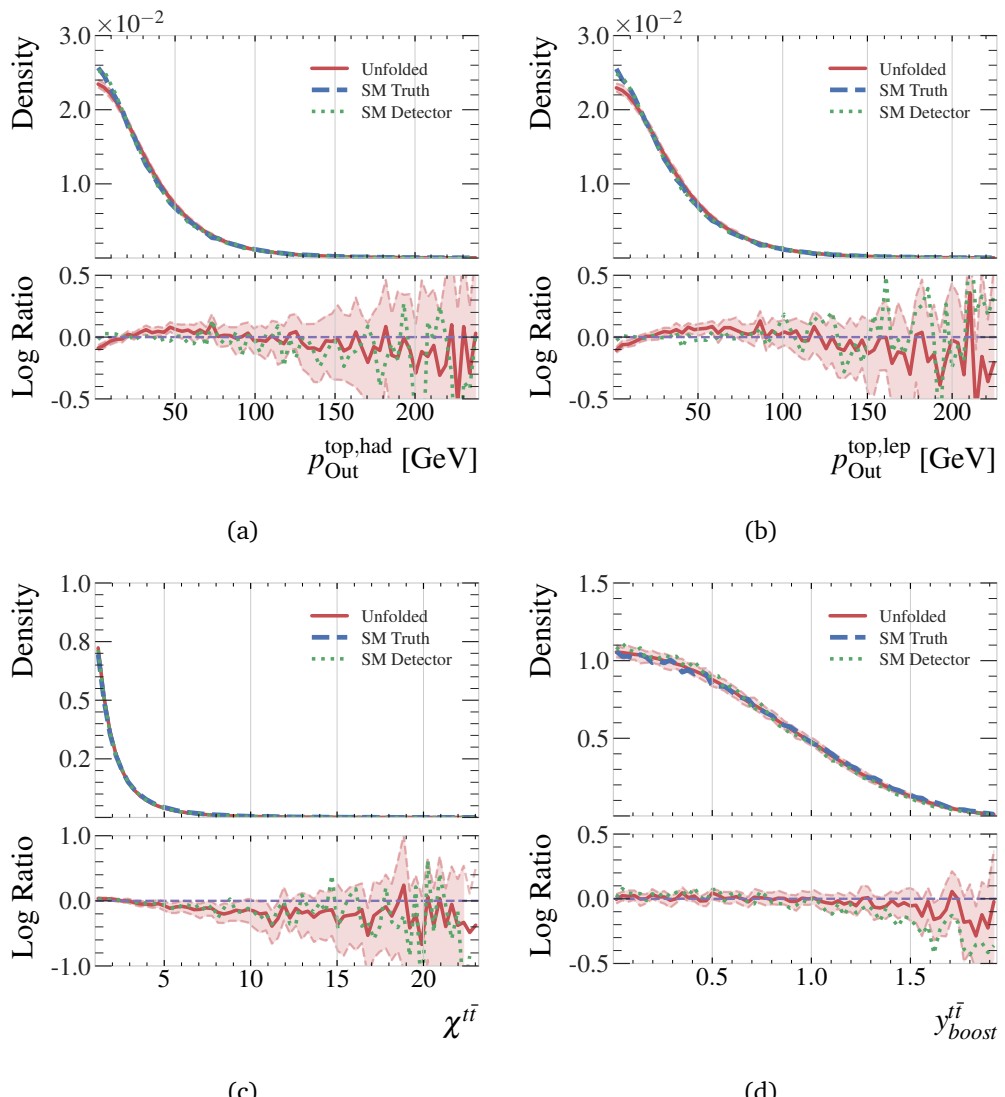

(a)

(b)

(c)

(d)

Figure 11: Distributions of reconstructed event-level observables related to the top quarks and $t\bar{t}$ system in the SM testing dataset. Shown are true particle-level (dashed blue), the unfolded particle-level (solid red), and the detector-level (dotted green). Unfolded distributions include error bounds estimated by sampling each event 128 times.

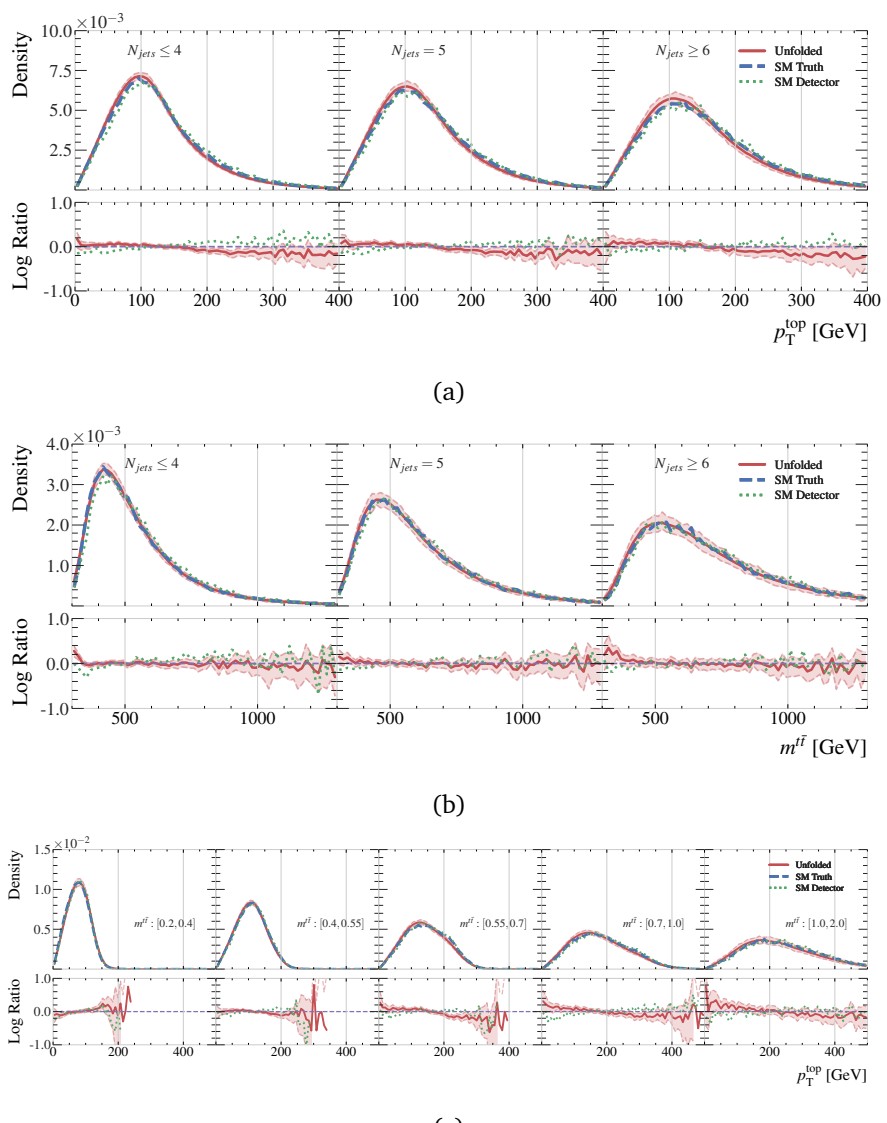

Figure 12: Distributions of (a) $p_T$ of the top quarks and (b) mass of the $t\bar{t}$ system in bins of jet multiplicity. Distributions of the $p_T$ of the top quarks in bins of $t\bar{t}$ mass is shown in (c). Shown are true particle-level (dashed blue), the unfolded particle-level (solid red), and the detector-level (dotted green). Unfolded distributions include error bounds estimated by sampling each event 128 times.

## 4.3 Performance on Dataset with BSM Physics Injection

To estimate the extent to which the posterior density modeled by VLD, $p(x|y)$, is dependent on the prior used to construct the training set $f_{\text{truth}}$ in Equation 3, the VLD network trained on the SM dataset is evaluated on the alternative sample containing BSM physics parametrized with a non-zero EFT operator (described in Section 4.1). The alternative sample contains a modified top-gluon vertex, leading to differences in the kinematic distributions used as unfolding targets as well as in the reconstructed distributions related to the top quarks, and is therefore a good probe of potential bias in the unfolding model due to the choice of SM prior.

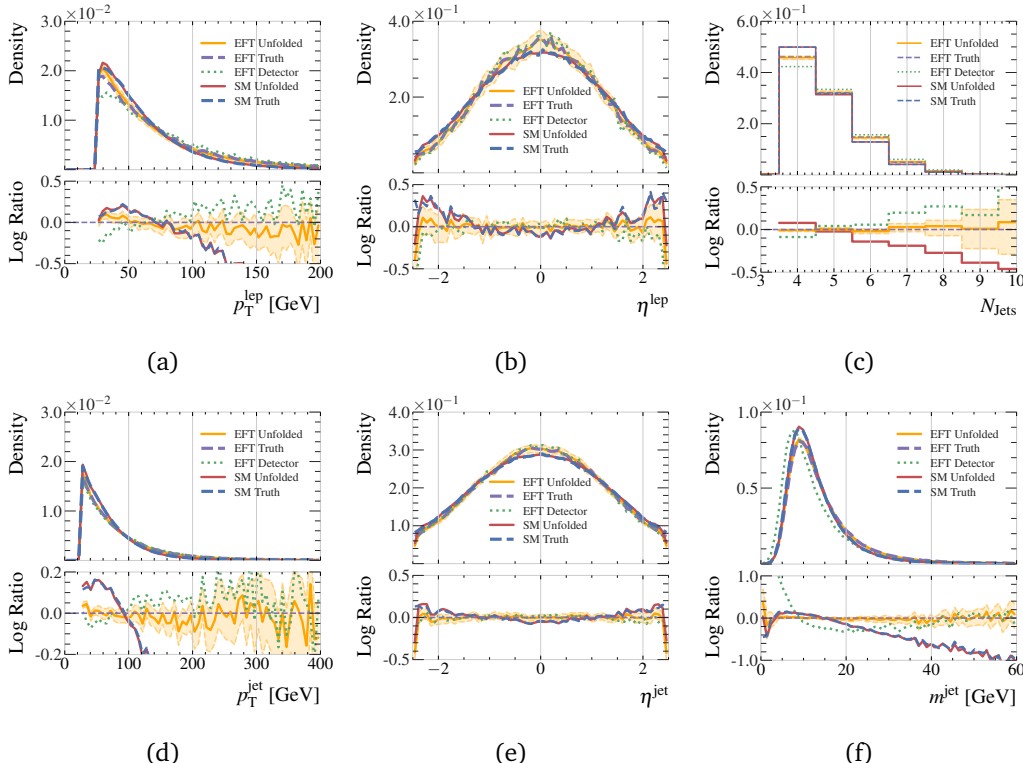

Figure 13: Distributions of lepton kinematic quantities in the EFT testing dataset, comparing the true particle-level leptons (dashed purple), the unfolded particle-level leptons (solid orange), and the detector-level leptons (dotted green). Also shown are the unfolded and truth SM distributions (solid red and dashed blue respectively). Unfolded distributions include error bounds estimated by sampling each event 128 times. The bottom pad shows the ratio with respect to the EFT truth distribution.

Figure 13 shows the truth, detector, and unfolded lepton and jet kinematics and multiplicity in the SM and EFT samples. There is good agreement almost everywhere between the unfolded and truth distributions for the EFT sample despite large differences with the SM training sample. The reconstructed top quark and $t\bar{t}$ system kinematics are shown in Fig. 14. They show similar disagreements in top $p_T$ and mass as was observed in unfolding the SM sample, which are much smaller than the differences between the SM and EFT truth distributions. This indicates that the choice of prior training sample has not induced a significant bias in the unfolding derived using VLD, at least in this test case. Nonetheless, it may be necessary to apply the iterative method proposed in [14] to remove all dependence on the prior when applying to real data. A full set of comparisons between the EFT and SM samples can be found in Appendix B.

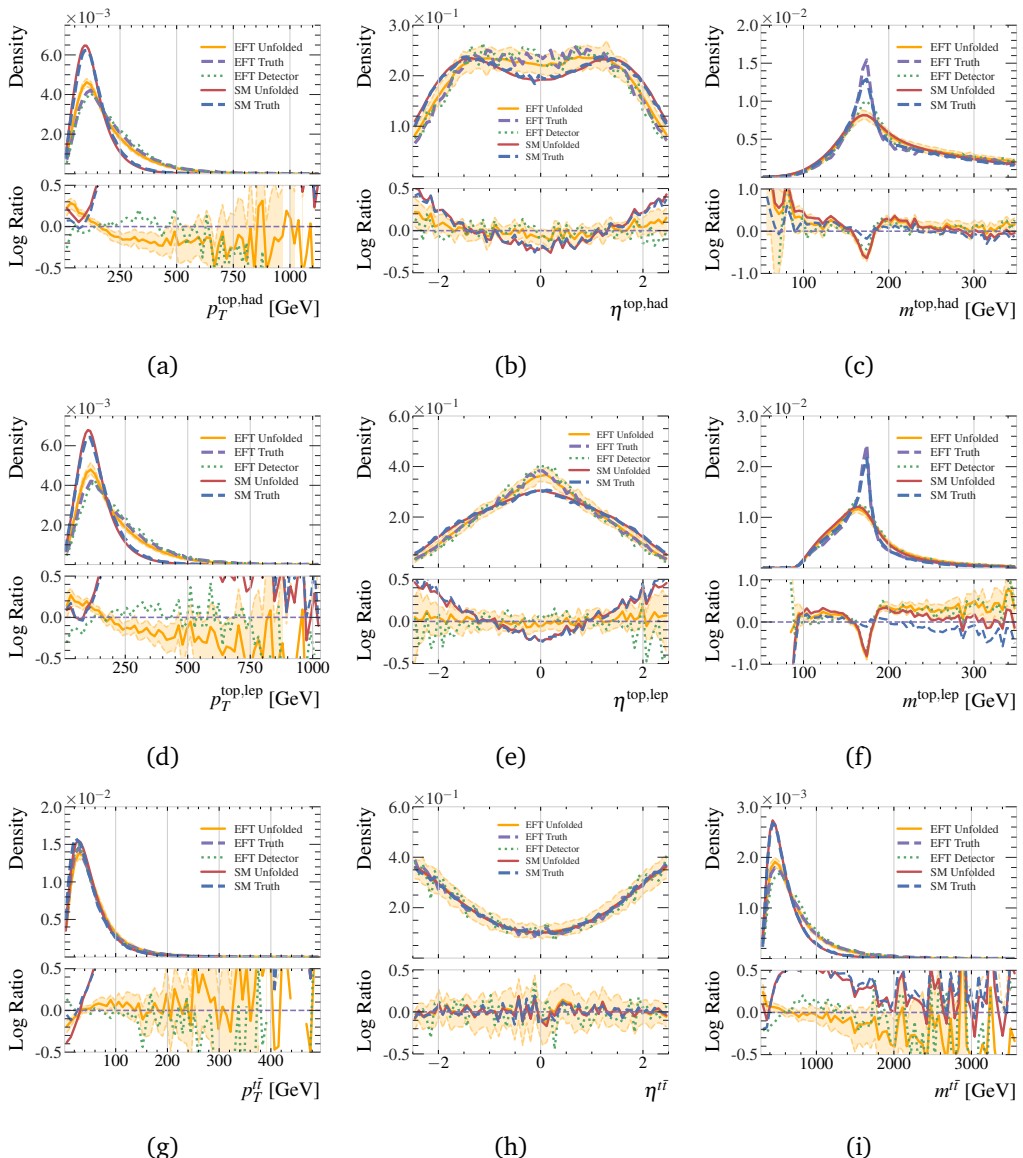

Figure 14: Reconstructed top quark and $t\bar{t}$ system kinematics in the EFT testing dataset. Shown are true particle-level (dashed purple), the unfolded particle-level (solid orange), and the detector-level (dotted green). Also shown are the unfolded and truth SM distributions (solid red and dashed blue respectively). Unfolded distributions include error bounds estimated by sampling each event 128 times. The bottom pad shows the ratio with respect to the EFT truth distribution.

# 5   Outlook and Discussion

This paper demonstrates the first application of generative ML models to full-event unfolding at particle-level, which requires a mechanism for unfolding the variable number of particles produced in collision events. No performance comparison to other generative approaches is provided, as none are yet capable of variable-length full-event unfolding. A comparison of VLD to the discriminative Omnifold method [10] on a variable- and high-dimensional unfolding task is left to future work.

In this paper, all final state particles not identified as leptons were clustered into jets before unfolding. A more ambitious approach to full-event unfolding would be to instead unfold before clustering. While the precise modeling of $\mathcal{O}(100)$ final state particles presents a greater challenge, there is no fundamental reason that the VLD approach would not scale to such a task. Exploring this possibility is also left to future work.

VLD was able to accurately predict truth-level distributions in the vast majority of phase space. However, mis-modelling was observed in regions where the truth-level distributions had sharp features or limited statistics in the training data. Previous work [18] accounted for the sharply peaked partonic top-quark mass distributions using a physics-inspired consistency loss term that ensured the predicted 4-vector components maintained the correct correlations. Here, this loss was not necessary to ensure consistency in the predicted 4-vectors for the particle-level leptons and jets. Mis-modelling was however observed in the sharply peaked particle-level top quark mass distributions. Since the particle-level top quark masses are reconstructed post-unfolding and VLD is not directly optimized to generate these distributions, applying a similar physics consistency loss is non-trivial. Other physics constraints, such as those enforcing symmetries or conversation laws, could also be incorporated into the objective in principle. Such physics constraints can easily be added to the loss function used to optimize the VAE; however adding such loss terms to the diffusion process would require performing computationally expensive inference at training time. Table 3 shows the distance metrics between detector and unfolded distributions for a selection of observables, distinguishing between the contributions to post-unfolding distance from the VAE or the diffusion parts of the network. The majority of the error comes from the diffusion process, and so an investigation of how best to utilize physics constraints is left to future work. Full error breakdown tables can be found in Appendix C.

There are several possibilities to improve performance where the truth-level distributions have few training samples. One is to simply avoid these regions, constructing the training sample with boundaries sufficiently far from the region to be unfolded. Another is to increase the number of training samples in these regions, as long as the unfolding remains insensitive to the prior distribution.

The presence of background events, which may complicate the unfolding, is not treated here. These background events are typically estimated using simulated data, and then subtracted before the unfolding step of the analysis. This subtraction could be achieved in an unbinned fashion with the addition of negative-weighted simulated background events in the training samples. These can be used directly in the training of the generative model [71], or after re-weighting the training sample to positive event weights [72].

Additionally the samples of simulated events used in realistic physics analyses typically contain events that pass one of the particle level or detector level event selections but not both. Events that pass the detector-level event selection but not the particle-level event selection (often called *fakes*) can be accounted for by simply including these events in the training data sets. At inference time, the events which migrate out of the particle level event selection are then not included in the final unfolded dataset. Events which pass the particle level event selection but not the detector level event selection (often called *inefficiencies*) are more problematic, since there is no detector level event on which to condition the model. The number of these

Table 3: An examination of the source for the error in unfolding. Presented are the distribution-free distance metrics from particle-level truth for the detector-level and unfolded distributions. The distances for the unfolded distributions are further subdivided into the distance produced by the VAE, evaluated by encoding and decoding the particle-level truth events and comparing the resulting distributions to particle-level truth, as well as the distance produced by the diffusion process, evaluated by calculating the distance metrics in the latent space of the VAE.

| Observable | Stage | Wasserstein | Energy | KL |
|---|---|---|---|---|
| $p_T^{Jet}$ | Detector | 2.30 | 0.27 | 0.26 |
| | Unfolding | $0.45 \pm 0.10$ | $0.05 \pm 0.01$ | $0.03 \pm 0.01$ |
| | VAE | $0.05 \pm 0.01$ | $0.01 \pm 0.00$ | $0.01 \pm 0.00$ |
| | Diffusion | $0.40 \pm 0.09$ | $0.04 \pm 0.01$ | $0.02 \pm 0.00$ |
| $E_T^{miss}$ | Detector | 3.03 | 0.41 | 0.95 |
| | Unfolding | $0.31 \pm 0.06$ | $0.04 \pm 0.01$ | $0.04 \pm 0.01$ |
| | VAE | $0.05 \pm 0.01$ | $0.01 \pm 0.00$ | $0.00 \pm 0.00$ |
| | Diffusion | $0.26 \pm 0.05$ | $0.04 \pm 0.01$ | $0.04 \pm 0.00$ |
| $H_T$ | Detector | 8.54 | 0.59 | 0.20 |
| | Unfolding | $7.45 \pm 0.54$ | $0.51 \pm 0.03$ | $0.20 \pm 0.02$ |
| | VAE | $1.67 \pm 0.12$ | $0.12 \pm 0.01$ | $0.03 \pm 0.01$ |
| | Diffusion | $5.78 \pm 0.43$ | $0.39 \pm 0.02$ | $0.17 \pm 0.02$ |
| $m^{top,lep}$ | Detector | 4.64 | 0.61 | 8.25 |
| | Unfolding | $5.03 \pm 0.20$ | $0.61 \pm 0.02$ | $9.16 \pm 0.44$ |
| | VAE | $1.13 \pm 0.08$ | $0.13 \pm 0.01$ | $0.53 \pm 0.04$ |
| | Diffusion | $3.90 \pm 0.13$ | $0.48 \pm 0.01$ | $8.63 \pm 0.40$ |
| $p_T^{top,had}$ | Detector | 4.25 | 0.39 | 0.25 |
| | Unfolding | $5.62 \pm 0.39$ | $0.48 \pm 0.03$ | $0.33 \pm 0.05$ |
| | VAE | $1.65 \pm 0.13$ | $0.15 \pm 0.01$ | $0.08 \pm 0.02$ |
| | Diffusion | $3.97 \pm 0.26$ | $0.33 \pm 0.02$ | $0.25 \pm 0.02$ |
| $m^{t\bar{t}}$ | Detector | 16.11 | 0.92 | 0.15 |
| | Unfolding | $7.58 \pm 1.58$ | $0.33 \pm 0.08$ | $0.04 \pm 0.01$ |
| | VAE | $0.58 \pm 0.06$ | $0.03 \pm 0.00$ | $0.02 \pm 0.01$ |
| | Diffusion | $6.99 \pm 1.52$ | $0.30 \pm 0.07$ | $0.02 \pm 0.01$ |
| $p_{Out}^{top,lep}$ | Detector | 0.27 | 0.03 | 0.26 |
| | Unfolding | $1.02 \pm 0.15$ | $0.17 \pm 0.02$ | $0.57 \pm 0.12$ |
| | VAE | $0.13 \pm 0.03$ | $0.02 \pm 0.00$ | $0.25 \pm 0.05$ |
| | Diffusion | $0.89 \pm 0.11$ | $0.16 \pm 0.02$ | $0.32 \pm 0.06$ |

events could be reduced by defining a signal region at detector level that avoids regions of phase space with poor detector efficiency.

Finally, realistic physics analyses must also account for sources of statistical and systematic uncertainty. The statistical uncertainties resulting from the finite size of the training sets can be estimated using bootstrapping methods [73]. One approach to propagating systematic uncertainties is to parameterize the model on these uncertainties [74] and unfold the data for several assumed values of the nuisance parameter, perhaps including constraints from auxiliary measurements. Exploration of this possibility is left to future work.

# 6 Conclusions

This paper presents the first application of generative unfolding techniques to a variable-dimensional unfolding task. Several modifications to the original VLD model made it possible for the model to naturally accommodate the variable-dimensional nature of the task of unfolding full experimental particle physics events.

In general, there is excellent closure between truth and unfolded distributions, both for a sample similar to the training sample, and an alternative. This lack of prior dependence strongly motivates the use of VLD for unfolding.

Some mis-modeling of kinematic distributions is seen near edges of the selection, due to a lack of training samples. Potential mitigation strategies were discussed in Section 5. Observables not included in the VLD training, such as the reconstructed top quark kinematics, showed larger mis-modeling. Most significant is the mis-modeling of the top quark mass distributions, which are sharply peaked at particle-level and not well reconstructed at detector level. Future work may incorporate additional physics constraints to better handle these kinds of observables.

The results demonstrate the possibility of performing full-event unfolding with a generative model, where the kinematics of all reconstructed jets, leptons, and the missing transverse momentum are unfolded to particle level, including handling a variable number of final state objects. Such measurements would be of great value to the high energy physics community, allowing results to be easily re-used and re-interpreted many years later, including the ability to construct event-level quantities not defined at the time of the original measurement. This would ensure a substantially longer lasting impact for all unfolded results that utilize VLD, and significantly reduce the number of individual efforts required to measure a large number of different observables.

## Acknowledgements

The authors thank Vinicius Mikuni, Ben Nachman, and Tilman Plehn for fruitful discussions on unfolding methods.

FUNDING INFORMATION    DW, KG, AG, and MF are supported by DOE grant DE-SC0009920. The work of AS and PB in part supported by ARO grant 76649-CS to PB.

# A  Variable Definitions

The observables displayed in Figures 11 and 19 are designed to be sensitive to various physics effects in $t\bar{t}$ production and decay. They were measured by the ATLAS collaboration [59] using traditional unfolding methods, and the distributions from this analysis are routinely used to test new theory predictions and tune relevant MC settings. These are included here to demonstrate the power of VLD unfolding, which would allow such variables (and any other that could be thought of) to be calculated post-unfolding. The definitions of these variables are as follows:

$$y^{t\bar{t}}{}_{\text{boost}} = \frac{1}{2}\left(y^{\text{top,had}} + y^{\text{top,lep}}\right) \tag{A.1}$$

$$y^* = \pm\frac{1}{2}\left(y^{\text{top,had}} - y^{\text{top,lep}}\right) \tag{A.2}$$

$$\chi^{t\bar{t}} = e^{2|y^*|} \tag{A.3}$$

$$p_{\text{out}}^{\text{top,had}} = \vec{p}^{\text{ top,had}} \cdot \frac{\vec{p}^{\text{ top,lep}} \times \vec{e}_z}{|\vec{p}^{\text{ top,lep}} \times \vec{e}_z|}, \tag{A.4}$$

$$p_{\text{out}}^{\text{top,lep}} = \vec{p}^{\text{ top,lep}} \cdot \frac{\vec{p}^{\text{ top,had}} \times \vec{e}_z}{|\vec{p}^{\text{ top,had}} \times \vec{e}_z|} \tag{A.5}$$

# B  Full set of EFT Distributions

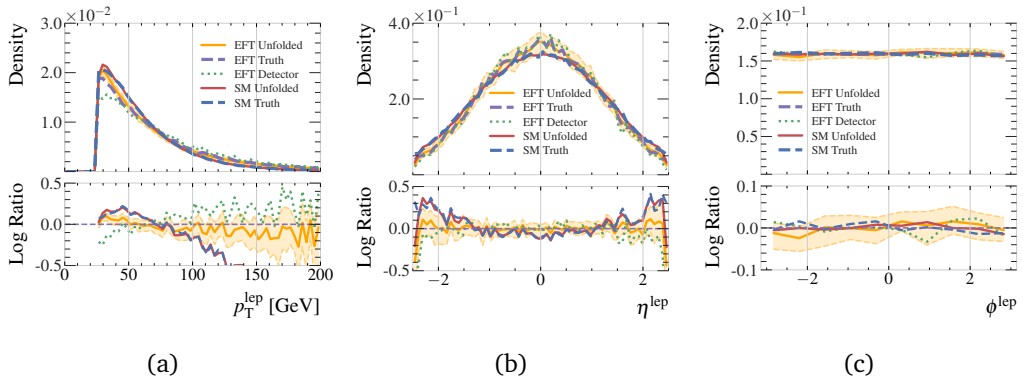

Figure 15: Kinematic distributions of leptons in the EFT testing dataset, comparing the true particle-level leptons (dashed purple), the unfolded particle-level leptons (solid orange), and the detector-level leptons (dotted green). Also shown are the unfolded and truth SM distributions (solid red and dashed blue respectively). Unfolded distributions include error bounds estimated by sampling each event 128 times. The bottom pad shows the ratio with respect to the EFT truth distribution.

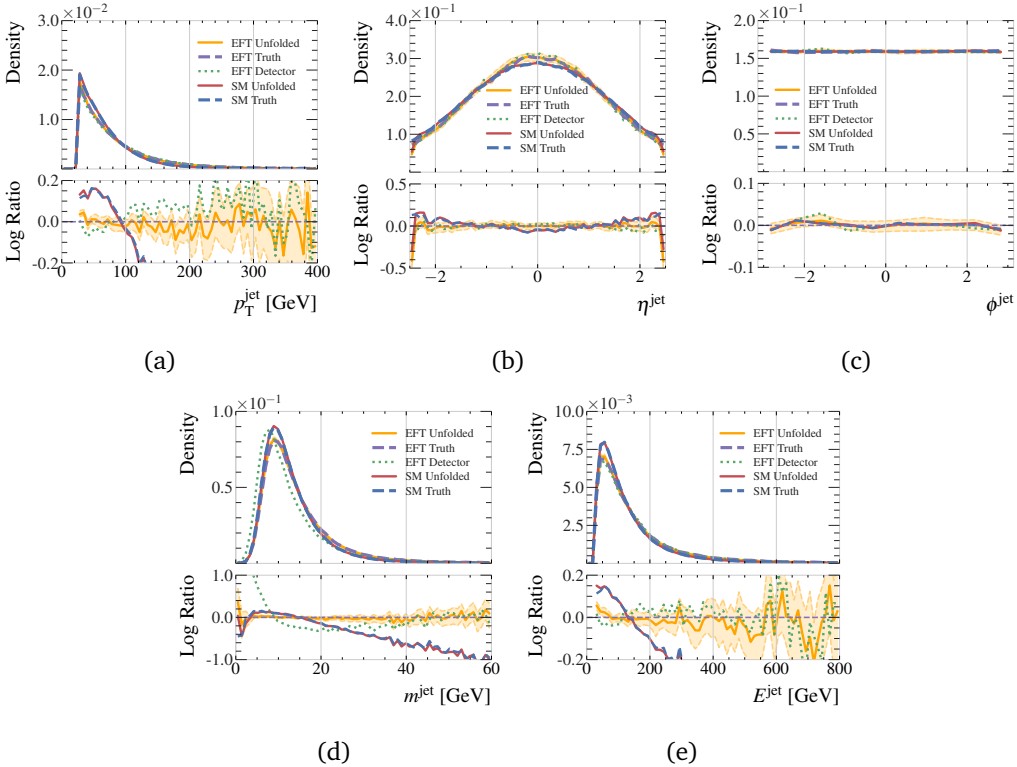

Figure 16: Kinematic distributions of all hadronic jets in the EFT testing dataset, comparing the true particle-level jets (dashed purple), the unfolded particle-level jets (solid orange), and the detector-level jets (dotted green). Also shown are the unfolded and truth SM distributions (solid red and dashed blue respectively). Unfolded distributions include error bounds estimated by sampling each event 128 times. The bottom pad shows the ratio with respect to the EFT truth distribution.

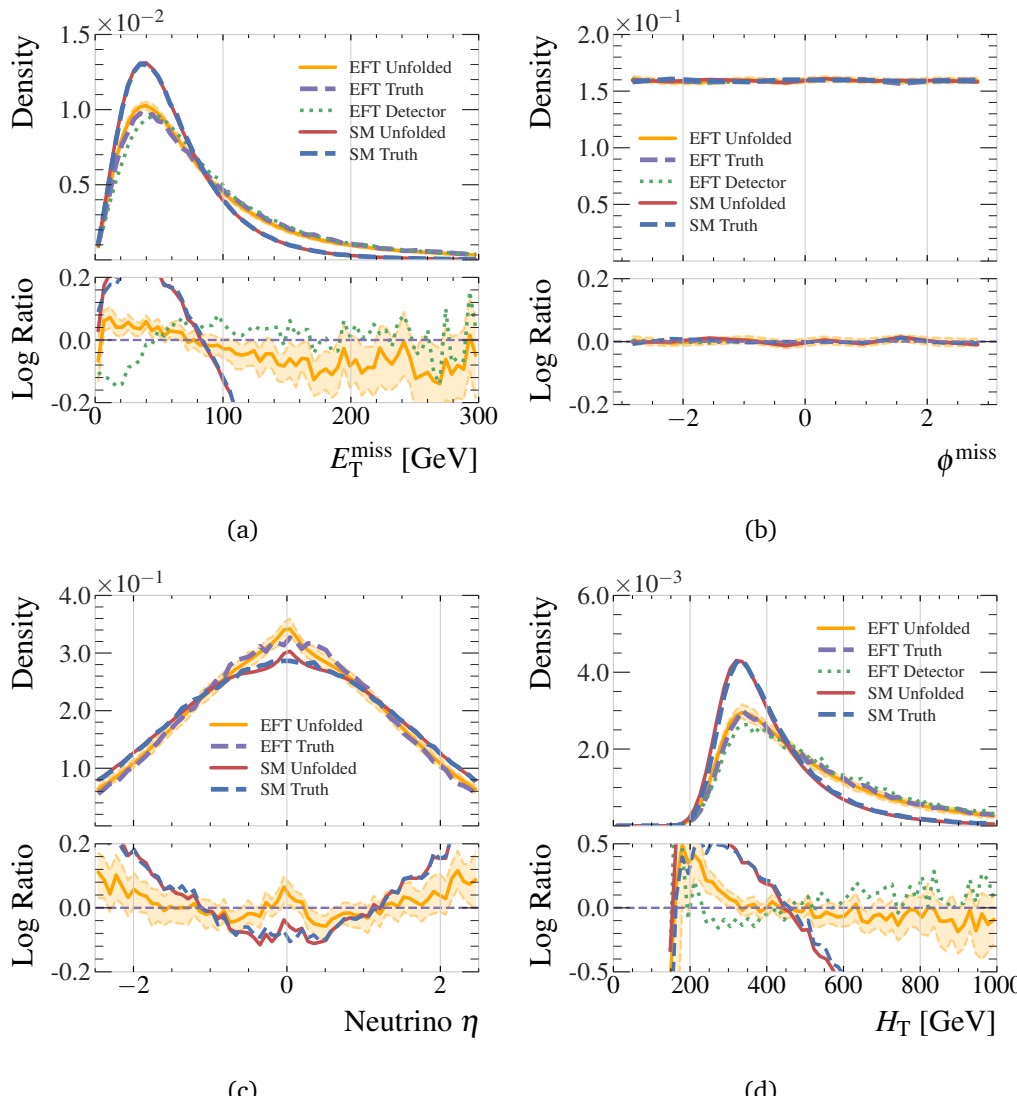

Figure 17: Event-level quantities in the EFT testing dataset, comparing the true particle-level events (dashed purple), the unfolded particle-level events (solid orange), and the detector-level events (dotted green). Also shown are the unfolded and truth SM distributions (solid red and dashed blue respectively). Unfolded distributions include error bounds estimated by sampling each event 128 times. The bottom pad shows the ratio with respect to the EFT truth distribution.

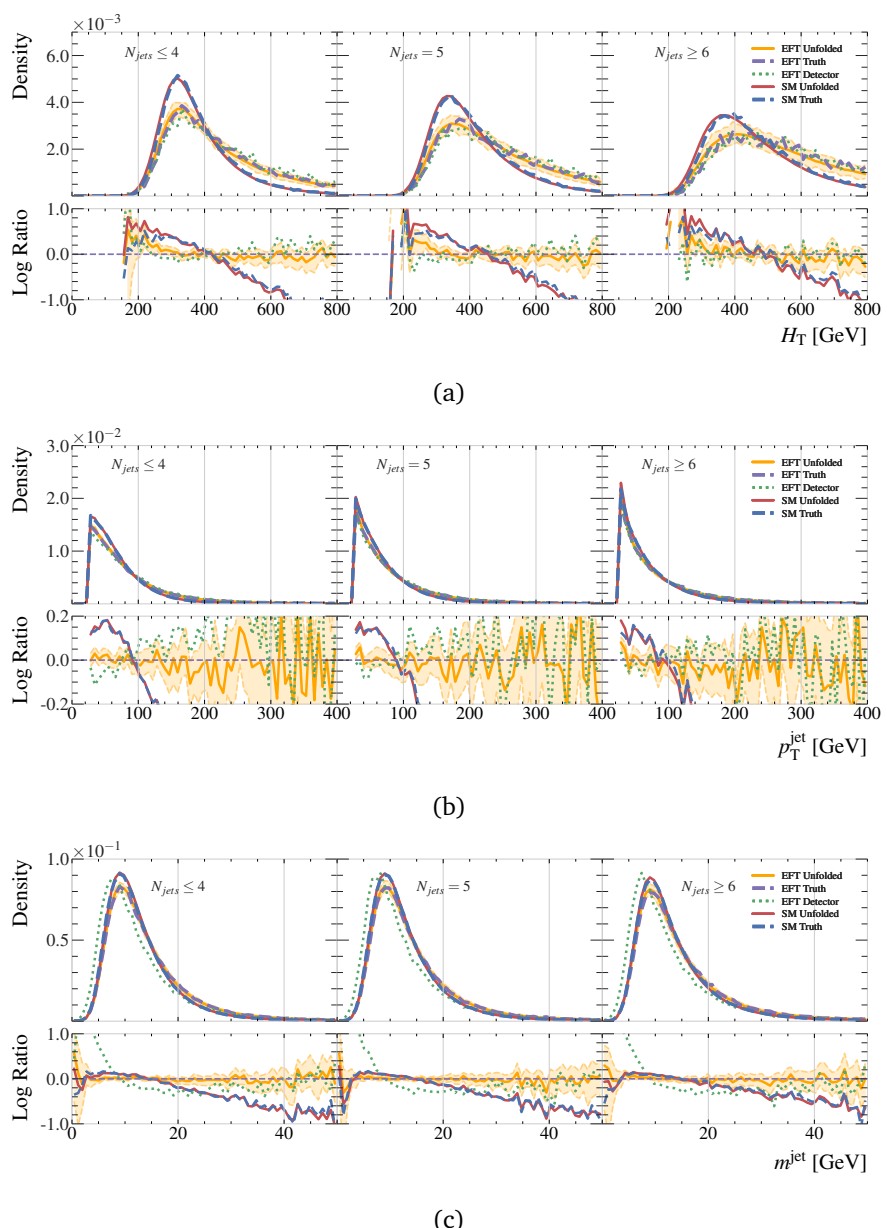

(a)

(b)

(c)

Figure 18: The event-level observable $H_T$, and the inclusive jet $p_T$ and mass, in the EFT testing dataset and binned by jet multiplicity. Shown are true particle-level jets (dashed purple), the unfolded particle-level jets (solid orange), and the detector-level jets (dotted green). Also shown are the unfolded and truth SM distributions (solid red and dashed blue respectively). Unfolded distributions include error bounds estimated by sampling each event 128 times. The bottom pad shows the ratio with respect to the EFT truth distribution.

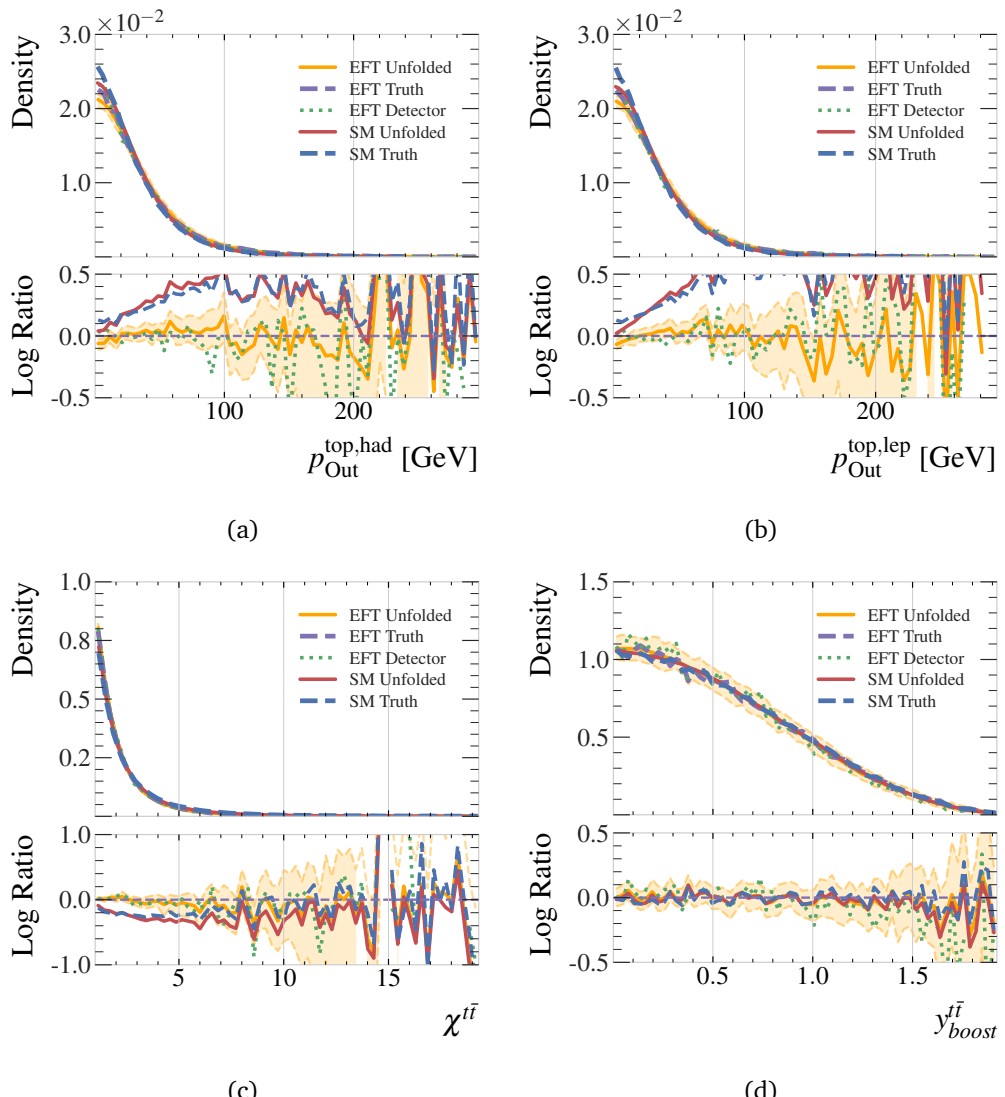

Figure 19: Reconstructed high-level observables related to the top quarks and $t\bar{t}$ system in the EFT testing dataset. Shown are true particle-level (dashed purple), the unfolded particle-level (solid orange), and the detector-level jets (dotted green). Also shown are the unfolded and truth SM distributions (solid red and dashed blue respectively). Unfolded distributions include error bounds estimated by sampling each event 128 times. The bottom pad shows the ratio with respect to the EFT truth distribution.

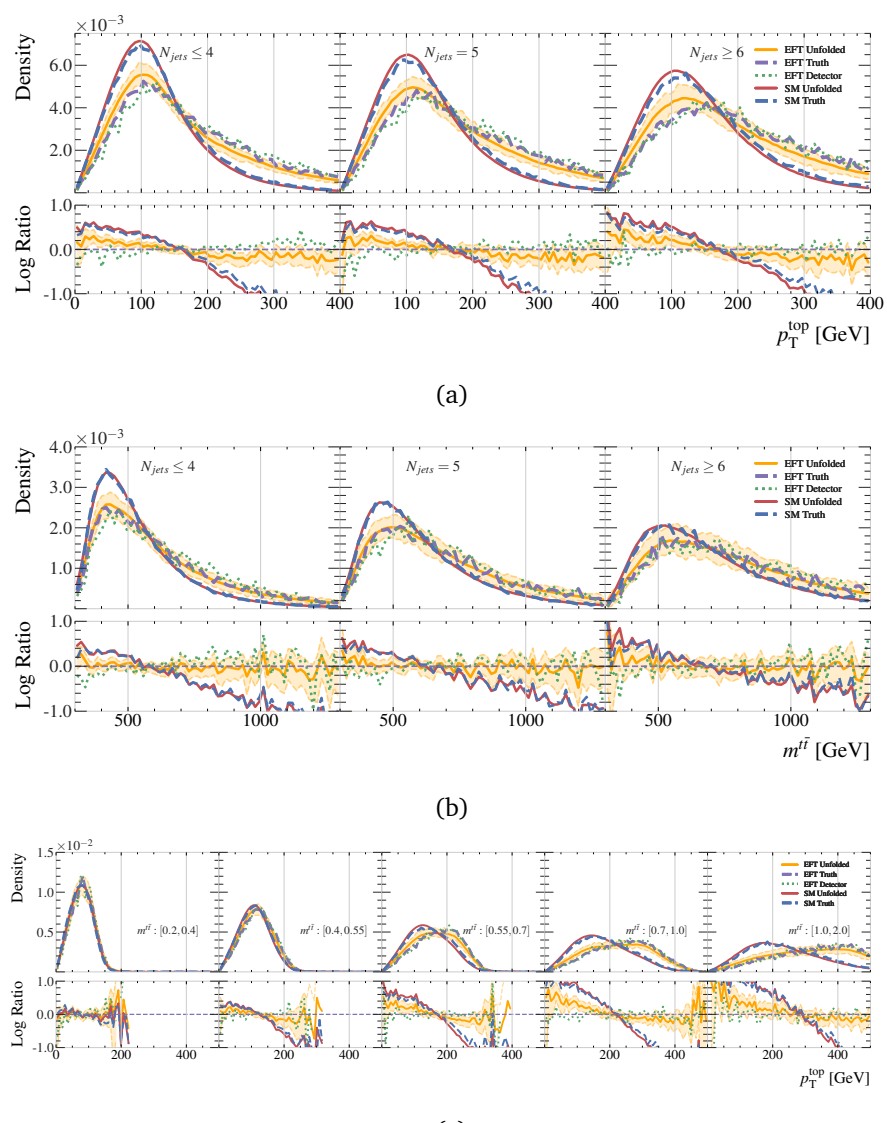

(a)

(b)

(c)

Figure 20: Distributions of the $p_T$ of the top quarks and mass of the $t\bar{t}$ system in bins of jet multiplicity are shown in (a) and (b). The distribution of the $p_T$ of the top quarks in bins of $t\bar{t}$ mass is shown in (c). The true particle-level (dashed purple), the unfolded particle-level (solid orange), and the detector-level (dotted green) distributions from the EFT dataset are shown, as are the the unfolded and truth SM distributions (solid red and dashed blue respectively). Unfolded distributions include error bounds estimated by sampling each event 128 times. The bottom pad shows the ratio with respect to the EFT truth distribution.

## C   Full Error Breakdown Tables

Table 4: An examination of the source for the error in unfolding. Presented are the distribution-free distance metrics from particle-level truth for the detector-level and unfolded distributions. The distances for the unfolded distributions are further subdivided into the distance produced by the VAE, evaluated by encoding and decoding the particle-level truth events and comparing the resulting distributions to particle-level truth, as well as the distance produced by the diffusion process, evaluated by calculating the distance metrics in the latent space of the VAE.

| Observable | Stage | Wasserstein | Energy | KL |
|---|---|---|---|---|
| $p_{\mathrm{T}}^{\mathrm{Jet}}$ | Detector | 2.30 | 0.27 | 0.26 |
| | Unfolding | $0.45 \pm 0.10$ | $0.05 \pm 0.01$ | $0.03 \pm 0.01$ |
| | VAE | $0.05 \pm 0.01$ | $0.01 \pm 0.00$ | $0.01 \pm 0.00$ |
| | Diffusion | $0.40 \pm 0.09$ | $0.04 \pm 0.01$ | $0.02 \pm 0.00$ |
| $\eta^{\mathrm{Jet}}$ | Detector | 0.01 | 0.01 | 3.44 |
| | Unfolding | $0.01 \pm 0.00$ | $0.00 \pm 0.00$ | $13.03 \pm 1.18$ |
| | VAE | $0.00 \pm 0.00$ | $0.00 \pm 0.00$ | $2.51 \pm 0.37$ |
| | Diffusion | $0.00 \pm 0.00$ | $0.00 \pm 0.00$ | $10.52 \pm 0.81$ |
| $\phi^{\mathrm{Jet}}$ | Detector | 0.00 | 0.00 | 0.02 |
| | Unfolding | $0.00 \pm 0.00$ | $0.00 \pm 0.00$ | $0.01 \pm 0.01$ |
| | VAE | $0.00 \pm 0.00$ | $0.00 \pm 0.00$ | $0.00 \pm 0.01$ |
| | Diffusion | $0.00 \pm 0.00$ | $0.00 \pm 0.00$ | $0.01 \pm 0.01$ |
| $m^{\mathrm{Jet}}$ | Detector | 1.52 | 0.52 | 61.66 |
| | Unfolding | $0.03 \pm 0.01$ | $0.01 \pm 0.00$ | $0.17 \pm 0.05$ |
| | VAE | $0.02 \pm 0.00$ | $0.01 \pm 0.00$ | $0.19 \pm 0.03$ |
| | Diffusion | $0.01 \pm 0.01$ | $0.00 \pm 0.00$ | $-0.02 \pm 0.02$ |
| $E^{\mathrm{Jet}}$ | Detector | 2.05 | 0.20 | 0.06 |
| | Unfolding | $0.66 \pm 0.26$ | $0.05 \pm 0.02$ | $0.01 \pm 0.00$ |
| | VAE | $0.07 \pm 0.00$ | $0.01 \pm 0.00$ | $0.01 \pm 0.01$ |
| | Diffusion | $0.59 \pm 0.25$ | $0.04 \pm 0.01$ | $0.00 \pm 0.01$ |
| $p_{\mathrm{T}}^{\mathrm{Lepton}}$ | Detector | 3.89 | 0.53 | 2.64 |
| | Unfolding | $0.27 \pm 0.10$ | $0.05 \pm 0.02$ | $0.17 \pm 0.05$ |
| | VAE | $0.07 \pm 0.00$ | $0.02 \pm 0.00$ | $0.14 \pm 0.04$ |
| | Diffusion | $0.20 \pm 0.10$ | $0.03 \pm 0.01$ | $0.03 \pm 0.01$ |
| $\eta^{\mathrm{Lepton}}$ | Detector | 0.03 | 0.02 | 56.01 |
| | Unfolding | $0.01 \pm 0.00$ | $0.01 \pm 0.00$ | $16.72 \pm 3.27$ |
| | VAE | $0.00 \pm 0.00$ | $0.00 \pm 0.00$ | $8.19 \pm 1.65$ |
| | Diffusion | $0.00 \pm 0.00$ | $0.00 \pm 0.00$ | $8.53 \pm 1.62$ |
| $\phi^{\mathrm{Lepton}}$ | Detector | 0.01 | 0.01 | 0.02 |
| | Unfolding | $0.01 \pm 0.01$ | $0.01 \pm 0.00$ | $0.08 \pm 0.05$ |
| | VAE | $0.00 \pm 0.00$ | $0.00 \pm 0.00$ | $0.01 \pm 0.01$ |
| | Diffusion | $0.01 \pm 0.00$ | $0.01 \pm 0.00$ | $0.07 \pm 0.04$ |

Table 5: An examination of the source for the error in unfolding. Presented are the distribution-free distance metrics from particle-level truth for the detector-level and unfolded distributions. The distances for the unfolded distributions are further subdivided into the distance produced by the VAE, evaluated by encoding and decoding the particle-level truth events and comparing the resulting distributions to particle-level truth, as well as the distance produced by the diffusion process, evaluated by calculating the distance metrics in the latent space of the VAE.

| Observable | Stage | Wasserstein | Energy | KL |
|---|---|---|---|---|
| $p_\mathrm{T}^\mathrm{top,had}$ | Detector | 4.25 | 0.39 | 0.25 |
| | Unfolding | $5.62 \pm 0.39$ | $0.48 \pm 0.03$ | $0.33 \pm 0.05$ |
| | VAE | $1.65 \pm 0.13$ | $0.15 \pm 0.01$ | $0.08 \pm 0.02$ |
| | Diffusion | $3.97 \pm 0.26$ | $0.33 \pm 0.02$ | $0.25 \pm 0.02$ |
| $\eta^\mathrm{top,had}$ | Detector | 0.01 | 0.01 | 8.40 |
| | Unfolding | $0.01 \pm 0.00$ | $0.01 \pm 0.00$ | $6.70 \pm 2.46$ |
| | VAE | $0.01 \pm 0.00$ | $0.00 \pm 0.00$ | $8.47 \pm 1.35$ |
| | Diffusion | $0.01 \pm 0.00$ | $0.00 \pm 0.00$ | $-1.77 \pm 1.11$ |
| $\phi^\mathrm{top,had}$ | Detector | 0.01 | 0.01 | 5.44 |
| | Unfolding | $0.01 \pm 0.01$ | $0.00 \pm 0.00$ | $3.76 \pm 1.35$ |
| | VAE | $0.00 \pm 0.00$ | $0.00 \pm 0.00$ | $4.23 \pm 1.08$ |
| | Diffusion | $0.00 \pm 0.00$ | $0.00 \pm 0.00$ | $-0.47 \pm 0.27$ |
| $E^\mathrm{top,had}$ | Detector | 10.92 | 0.60 | 0.08 |
| | Unfolding | $2.50 \pm 1.14$ | $0.10 \pm 0.05$ | $0.02 \pm 0.01$ |
| | VAE | $0.52 \pm 0.08$ | $0.03 \pm 0.01$ | $0.01 \pm 0.00$ |
| | Diffusion | $1.98 \pm 1.06$ | $0.08 \pm 0.05$ | $0.01 \pm 0.01$ |
| $m^\mathrm{top,had}$ | Detector | 4.05 | 0.50 | 2.37 |
| | Unfolding | $4.62 \pm 0.32$ | $0.55 \pm 0.04$ | $4.45 \pm 0.26$ |
| | VAE | $1.24 \pm 0.05$ | $0.15 \pm 0.01$ | $0.43 \pm 0.04$ |
| | Diffusion | $3.37 \pm 0.26$ | $0.40 \pm 0.02$ | $4.02 \pm 0.22$ |
| $p_\mathrm{T}^\mathrm{top,lep}$ | Detector | 7.48 | 0.66 | 0.66 |
| | Unfolding | $4.59 \pm 0.41$ | $0.41 \pm 0.04$ | $0.32 \pm 0.05$ |
| | VAE | $1.14 \pm 0.08$ | $0.11 \pm 0.01$ | $0.07 \pm 0.02$ |
| | Diffusion | $3.45 \pm 0.32$ | $0.31 \pm 0.03$ | $0.25 \pm 0.03$ |
| $\eta^\mathrm{top,lep}$ | Detector | 0.05 | 0.04 | 51.12 |
| | Unfolding | $0.01 \pm 0.00$ | $0.01 \pm 0.00$ | $8.29 \pm 3.34$ |
| | VAE | $0.00 \pm 0.00$ | $0.00 \pm 0.00$ | $10.79 \pm 1.05$ |
| | Diffusion | $0.01 \pm 0.00$ | $0.00 \pm 0.00$ | $-2.50 \pm 2.29$ |
| $\phi^\mathrm{top,lep}$ | Detector | 0.01 | 0.01 | 6.15 |
| | Unfolding | $0.01 \pm 0.01$ | $0.01 \pm 0.01$ | $4.89 \pm 1.84$ |
| | VAE | $0.01 \pm 0.00$ | $0.00 \pm 0.00$ | $4.32 \pm 0.81$ |
| | Diffusion | $0.01 \pm 0.01$ | $0.00 \pm 0.01$ | $0.57 \pm 1.03$ |
| $E^\mathrm{top,lep}$ | Detector | 4.71 | 0.32 | 0.13 |
| | Unfolding | $8.32 \pm 1.16$ | $0.45 \pm 0.06$ | $0.09 \pm 0.02$ |
| | VAE | $1.19 \pm 0.09$ | $0.07 \pm 0.00$ | $0.03 \pm 0.01$ |
| | Diffusion | $7.13 \pm 1.07$ | $0.39 \pm 0.06$ | $0.06 \pm 0.01$ |
| $m^\mathrm{top,lep}$ | Detector | 4.64 | 0.61 | 8.25 |
| | Unfolding | $5.03 \pm 0.20$ | $0.61 \pm 0.02$ | $9.16 \pm 0.44$ |
| | VAE | $1.13 \pm 0.08$ | $0.13 \pm 0.01$ | $0.53 \pm 0.04$ |
| | Diffusion | $3.90 \pm 0.13$ | $0.48 \pm 0.01$ | $8.63 \pm 0.40$ |

Table 6: An examination of the source for the error in unfolding. Presented are the distribution-free distance metrics from particle-level truth for the detector-level and unfolded distributions. The distances for the unfolded distributions are further subdivided into the distance produced by the VAE, evaluated by encoding and decoding the particle-level truth events and comparing the resulting distributions to particle-level truth, as well as the distance produced by the diffusion process, evaluated by calculating the distance metrics in the latent space of the VAE.

| Observable | Stage | Wasserstein | Energy | KL |
|---|---|---|---|---|
| $m^{t\bar{t}}$ | Detector | 16.11 | 0.92 | 0.15 |
| | Unfolding | $7.58 \pm 1.58$ | $0.33 \pm 0.08$ | $0.04 \pm 0.01$ |
| | VAE | $0.58 \pm 0.06$ | $0.03 \pm 0.00$ | $0.02 \pm 0.01$ |
| | Diffusion | $6.99 \pm 1.52$ | $0.30 \pm 0.07$ | $0.02 \pm 0.01$ |
| $\chi^{t\bar{t}}$ | Detector | 0.09 | 0.04 | 4.52 |
| | Unfolding | $0.11 \pm 0.02$ | $0.05 \pm 0.01$ | $5.41 \pm 1.47$ |
| | VAE | $0.03 \pm 0.00$ | $0.02 \pm 0.00$ | $2.90 \pm 0.67$ |
| | Diffusion | $0.08 \pm 0.01$ | $0.04 \pm 0.00$ | $2.51 \pm 0.80$ |
| $y^{t\bar{t}}_{boost}$ | Detector | 0.02 | 0.03 | 80.18 |
| | Unfolding | $0.01 \pm 0.00$ | $0.01 \pm 0.00$ | $22.50 \pm 6.51$ |
| | VAE | $0.00 \pm 0.00$ | $0.00 \pm 0.00$ | $25.15 \pm 4.86$ |
| | Diffusion | $0.01 \pm 0.00$ | $0.01 \pm 0.00$ | $-2.65 \pm 1.65$ |
| $p^{top,had}_{Out}$ | Detector | 0.48 | 0.06 | 0.28 |
| | Unfolding | $0.82 \pm 0.15$ | $0.14 \pm 0.02$ | $0.42 \pm 0.10$ |
| | VAE | $0.17 \pm 0.05$ | $0.02 \pm 0.01$ | $0.30 \pm 0.02$ |
| | Diffusion | $0.65 \pm 0.10$ | $0.12 \pm 0.02$ | $0.12 \pm 0.07$ |
| $p^{top,lep}_{Out}$ | Detector | 0.27 | 0.03 | 0.26 |
| | Unfolding | $1.02 \pm 0.15$ | $0.17 \pm 0.02$ | $0.57 \pm 0.12$ |
| | VAE | $0.13 \pm 0.03$ | $0.02 \pm 0.00$ | $0.25 \pm 0.05$ |
| | Diffusion | $0.89 \pm 0.11$ | $0.16 \pm 0.02$ | $0.32 \pm 0.06$ |
| $E^{miss}_{T}$ | Detector | 3.03 | 0.41 | 0.95 |
| | Unfolding | $0.31 \pm 0.06$ | $0.04 \pm 0.01$ | $0.04 \pm 0.01$ |
| | VAE | $0.05 \pm 0.01$ | $0.01 \pm 0.00$ | $0.00 \pm 0.00$ |
| | Diffusion | $0.26 \pm 0.05$ | $0.04 \pm 0.01$ | $0.04 \pm 0.00$ |
| $H_{T}$ | Detector | 8.54 | 0.59 | 0.20 |
| | Unfolding | $7.45 \pm 0.54$ | $0.51 \pm 0.03$ | $0.20 \pm 0.02$ |
| | VAE | $1.67 \pm 0.12$ | $0.12 \pm 0.01$ | $0.03 \pm 0.01$ |
| | Diffusion | $5.78 \pm 0.43$ | $0.39 \pm 0.02$ | $0.17 \pm 0.02$ |

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
