# Peer review of "Full Event Particle-Level Unfolding with Variable-Length Latent Variational Diffusion"

_SciPost Physics_

## Round 1 · Referee Report · Anonymous (Referee 1) · 2024-6-7

Strengths
- The method and architecture is well described and a lot of details are given
- It fills a gap in the literature: it is the first method to unfold full-event (variable dimensional) collider data
Weaknesses
- one of the main results of the paper, that the unfolding performance is independent of the training data distribution, could be explained / investigated in more detail
Report
My biggest concern is about the observations in section 4.3 in combination with statement in section 6: "This lack of prior dependence strongly motivates the use of VLD for unfolding.". It is true that the model performs well on a different distribution as seen in training on the dataset considered here. But, it is not clear that this generalizes beyond this example. I understand that the authors cannot look at many processes within the scope of this manuscript, but I would at least like to see some explanation or investigation of why such a behavior would be expected. If the authors want to keep the sentence in the conclusion, they need to provide more explanantion / tests / examples to support the claim.
Apart from this, I have a few minor proposals that could enhance the manuscript, see the list below.
I therefore think that the manuscript should be published in SciPost Physics after the concerns have been addressed.
Requested changes
- One of the strengths of generative unfolding: event-by-event uncertainties, (since a given detector level input can be unfolded several times) could be mentioned in the introduction, since this an additional reason to use this method compared to discriminative methods.
- In section 3.1, is O_P_0 treated differently by the position-equivariant transformer with respect to the other O_P_i? Please explain.
- In section 3.3, why is y_0 needed? Please explain.
- In section 3.4, how is ensured that the ordering is constant over t? Or is that not a problem to be concerned of?
- In section 4.1, have both coordinate representations P^cart and P^Polar been used simultaneously (concatenated)?
- In sections 4.2 and 4.3, to better visualize the correlations between the observables and how well these are learned by the model, I suggest to add corner plots (for example to an appendix). Maybe this also explains the performance on down-stream observables a bit more.
- In sections 4.2 and 4.3, in addition to the metrics shown in the tables, I think it would be nice to see how well a neural classifier (see discussion in 2305.16774) would be able to distinguish unfolded from true events.
- In figures 5c and 7b (c), I suggest to zoom in on the bottom panel a bit.
- In the tables, I suggest to add a test of 'truth vs truth' to get a better feeling for the natural spread of the metrics. (i.e. is a distance of 0.04 a lot?)
- The authors refer a lot to reference 18, which is ok. However, it would be nice if the definitions of the metrics used in the tables could be replicated here as well and are not kept in Appendix C of Ref 18 only.
- In table 3, the errors of VAE and diffusion seem to add perfectly to the Unfolding error. Is that by construction or a non-trivial cross check on how the metrics are evaluated?
- Please make your code and training data available via git / zenodo / others.
Recommendation
Ask for minor revision
Dear reviewer, Thank you very much for the detailed and helpful review. We would like to apologise for the delay in resubmission. Our lead author was away on internship for the summer. We hope the new version of the paper addresses your concerns, and we have some responses for you below. Sincerely,
Kevin for the team "My biggest concern is about the observations in section 4.3 in combination with statement in section 6: "This lack of prior dependence strongly motivates the use of VLD for unfolding.". It is true that the model performs well on a different distribution as seen in training on the dataset considered here. But, it is not clear that this generalizes beyond this example. I understand that the authors cannot look at many processes within the scope of this manuscript, but I would at least like to see some explanation or investigation of why such a behavior would be expected. If the authors want to keep the sentence in the conclusion, they need to provide more explanantion / tests / examples to support the claim." Response - We agree with the reviewers' conclusions here. Though we show good performance on one alternative distribution, there are many other possible distributions we could consider, so we should not make broad claims about the level of prior dependence of this method. We have removed the relevant statement from the conclusion. "One of the strengths of generative unfolding: event-by-event uncertainties, (since a given detector level input can be unfolded several times) could be mentioned in the introduction, since this an additional reason to use this method compared to discriminative methods." Response- We do believe that the ability to unfold a single event multiple times is an additional benefit of generative models. We have added a sentence to this regard in Section 4.2, along with a citation to a paper that demonstrates how use of generative models can improve the statistical power of particle physics datasets. However we are not sure that “event-by-event uncertainties” have an application in a physics analysis, so we elected to leave this detail out. "In section 3.1, is O_P_0 treated differently by the position-equivariant transformer with respect to the other O_P_i? Please explain." Response- The transformer is position-equivariant with respect to the inputs. This means that if the O_P were shuffled, the outputs x would also shuffle in the same way. It’s then only important that the vector describing the event level quantities is given a fixed position, in this case the first one. This way we know what w to apply the event predictor to after the particle decoder (see Figure 1). "In section 3.3, why is y_0 needed? Please explain." Response - Y_0 is a learned vector that is added to the set of observables to serve as a carrier for the multiplicity information. Since the multiplicity depends on all of the inputs, but is itself a single value, we need an additional vector that extracts contextual information and feeds it to the multiplicity predictor. "In section 3.4, how is ensured that the ordering is constant over t? Or is that not a problem to be concerned of?" Response - The ordering is only enforced during network training in the diffusion loss term. Since we are training, we can make use of the truth-level information, so the objects are ordered by truth pT which doesn’t change at high-levels of noise. During inference, everything is fully equivariant. "In section 4.1, have both coordinate representations P^cart and P^Polar been used simultaneously (concatenated)?" Response - Yes, for the input to the detector encoder the representations are concatenated and used together. "In sections 4.2 and 4.3, to better visualize the correlations between the observables and how well these are learned by the model, I suggest to add corner plots (for example to an appendix). Maybe this also explains the performance on down-stream observables a bit more." Response - We have added corner plots to the appendix and they are quite interesting to look at, but we do not notice any large difference between the predicted and true correlations. The only visible differences between the predicted and true plots is in the top quark mass distributions, but these differences are already captured in Figure 10. The shapes of the rest of the unfolded distributions are very accurate. "In sections 4.2 and 4.3, in addition to the metrics shown in the tables, I think it would be nice to see how well a neural classifier (see discussion in 2305.16774) would be able to distinguish unfolded from true events." Response - We expect that a neural classifier would be able to distinguish unfolded from true easily, for example using the kinematics of the top quarks in Section 4.2 which are mismodeled. We agree that achieving the level of precision where the unfolded events are not distinguishable from the true events is an important milestone, but we are confident we are currently not at this precision. "In figures 5c and 7b (c), I suggest to zoom in on the bottom panel a bit." Response - We thank the reviewer for the suggestion and rescaled the log ratio plots across the paper to fit better for the observables. " In the tables, I suggest to add a test of 'truth vs truth' to get a better feeling for the natural spread of the metrics. (i.e. is a distance of 0.04 a lot?)" Response - Any difference from 0 in the distance metrics between truth and itself would only result from statistical uncertainties. With the statistics we have, these uncertainties are small enough that the metrics evaluate to 0 within numerical precision. We have added a sentence stating that the statistical uncertainties in these distributions is much smaller than the uncertainties obtained from sampling the generative model many times, in Section 4.2. "The authors refer a lot to reference 18, which is ok. However, it would be nice if the definitions of the metrics used in the tables could be replicated here as well and are not kept in Appendix C of Ref 18 only." Response - We agree with this suggestion and have added the definition of the metrics to Appendix B. Note this is simply copy / pasted from the appendix of Ref 18. "In table 3, the errors of VAE and diffusion seem to add perfectly to the Unfolding error. Is that by construction or a non-trivial cross check on how the metrics are evaluated?" Response - This is a slightly non-trivial cross check. Since our particle-level decoder is unconditional it can’t move the unfolded distribution any closer to the truth. Ideally it would add zero distance, but it adds some in practice. The sum of this distance, and the distance evaluated in the VAE latent space, is the total distance within the statistical errors. If you look closely there are some places where the VAE and diffusion errors do not add perfectly to the total, due to the statistical errors (e.g. bottom row energy distance in Table 3). "Please make your code and training data available via git / zenodo / others." Response - The code is available at https://github.com/Alexanders101/LVD/tree/main and the data at https://zenodo.org/records/13364827. We have also added these links to the paper.

Author: Kevin Greif on 2024-10-18 [id 4875]
(in reply to Report 2 on 2024-07-01)Dear reviewer,
Thank you very much for the detailed and helpful review. We would like to apologise for the delay in resubmission. Our lead author was away on internship for the summer. We hope the new version of the paper addresses your concerns, and we have some responses for you below.
Sincerely, Kevin for the team
All metrics considered by the authors (reported in Ref. [18], and that it would be useful to report here) seem to be sums of 1D distances computed on the 1D marginal distributions corresponding to the features of interest. Such metrics are therefore not naturally expected to be sensitive to correlations among features. I suggest the authors to consider adding at least one metric with some expected sensitivity on the correlation. One option, which does not require too intensive calculations and that is still based on 1D distances, could be the Sliced Wasserstein Distance.
Response - The top-quark kinematics presented in Figure 10 are finely dependent on the correlations present between the directly predicted features (the kinematics of the jets, leptons, and the missing transverse momentum). They are more dependent on the correlations than something like the sliced wasserstein, since they have a quadratic dependence on the directly predicted features instead of linear. Therefore we think it is enough to look at the 1D marginal distances in these particular distributions, especially because we see limited performance in the top mass distributions.
To better understand and visualize the performances beyond the 1D marginal distributions, on which evaluation metrics are computed, I would suggest the authors to add corner plots.
Response - We have added corner plots to the appendix and they are quite interesting to look at, but we do not notice any large difference between the predicted and true correlations. The only visible differences between the predicted and true plots is in the top quark mass distributions, but these differences are already captured in Figure 10. The shapes of the rest of the unfolded distributions are very accurate.
The code and dataset need to be made public (for instance on GitHub and Zenodo) to allow full reproducibility of the analysis and broader usage of the method.
Response - The code is available at https://github.com/Alexanders101/LVD/tree/main and the data at https://zenodo.org/records/13364827. We have also added this information to the paper.

---

## Round 1 · Referee Report · Anonymous (Referee 2) · 2024-7-1

Strengths
2. Comprehensive Evaluation: The performance of the method is thoroughly evaluated in the context of semi-leptonic top quark pair production at the LHC, providing clear and detailed results.
3. Addressing High-Dimensional Data: The method effectively addresses the challenge of unfolding unbinned distributions in high-dimensional spaces, which is a substantial improvement over traditional techniques.
4. Potential Impact: The proposed method pushes the edges of the performances of generative unfolding methods setting new standards for full-event unfolding.
Weaknesses
2. Computational Complexity: The training and inference processes for the VLD model are computationally intensive, which might limit its practical applicability for extremely large datasets or for use in environments with limited computational resources.
3. Specificity of top pair production: While the method is tested on semi-leptonic top quark pair production, its performance on other processes or in different experimental setups is not explored, leaving , as expected, questions about its generalizability.
4. Iterative Prior Adjustment: The paper mentions that the dependence on the training set prior might be mitigated through an iterative method, but this approach is not demonstrated in the current work. Further exploration and validation of this iterative adjustment would strengthen the findings.
Report
The comprehensive evaluation on semi-leptonic top pair production at the LHC provides convincing evidence of the effectiveness of the method, although the results feature areas of the distributions where the model struggles, particularly in regions with sparse data. The computational demands of the model and its performance on other types of data are also areas that could benefit from further exploration.
The paper certainly meets the criteria for acceptance in this journal. It presents original and significant contributions to the field, demonstrates clear and thorough methodology, and discusses the results and limitations transparently. The clarity of the writing is excellent.
Requested changes
1. All metrics considered by the authors (reported in Ref. [18], and that it would be useful to report here) seem to be sums of 1D distances computed on the 1D marginal distributions corresponding to the features of interest. Such metrics are therefore not naturally expected to be sensitive to correlations among features. I suggest the authors to consider adding at least one metric with some expected sensitivity on the correlation. One option, which does not require too intensive calculations and that is still based on 1D distances, could be the Sliced Wasserstein Distance.
2. To better understand and visualize the performances beyond the 1D marginal distributions, on which evaluation metrics are computed, I would suggest the authors to add corner plots.
3. The code and dataset need to be made public (for instance on GitHub and Zenodo) to allow full reproducibility of the analysis and broader usage of the method.
Recommendation
Ask for minor revision

---

## Round 2 · Referee Report · Anonymous (Referee 3) · 2024-12-17

Strengths
Making the VLD approach to unfolding flexible enough to accommodate unfolding problems with varying dimensionality is an excellent proposal.
Moreover the paper is well structured and clearly written.
Weaknesses
The description of the algorithm as well as some arguments sometimes lack clarity.
Report
Once the requested changes are addressed I recommend the paper for publication.
Requested changes
1. Typos and similar
- "They have been been"
- "D_{DENOISE}, This"
- Eq. 10/11 the tilde is off.
- Fig.3 should include \bar{t}, \bar{b}etc.
- Fig. 5: Some labels (a,b,c) are only half visible.
- Fig .13 c seems to be missing the SM truth line?
- empty page 33
2. You state "Because generative approaches require only synthetic data during training, they do not suffer from this limitation." This is only half correct. Indeed the initial unfolding algorithm can be trained on unlimited statistics of the MC. Once you observe a prior dependence are you have to iterate, the limited amount of available data does become a limiting factor for the iteration. Maybe it is possible to argue that the unfolding is still less affected though.
3. Fig. 1: This figure is very hard to follow. While it becomes clearer when reading the paper, a few design choices could be altered to ensure better readability.
1. Fix a main direction that conveys the main task you want to illustrate.
2. Avoid edges when possible, even if the figure becomes bigger.
3. The execution of the unfolding should lead to an arrow from (x_0,.., x_N) to the particle decoder, which I would consider the main direction. However that direction is not indicated.
4. Related to Fig. 1 and Fig. 2 several choices are not clear:
1. The role of $y_0$ as a learnable parameter. It seems to be independent from any input. So it is a learnable constant? Why is this not covered by the Multiplicity Predictor itself?
2. Is the sampling of the latent space in the VAE considered part of the encoder or the decoder and at which part does the output of the Denoising Network enter?
3. Is the transformer encoder in Fig. 2 an encoder or the denoising network in Fig. 1?
4. The relative factor between the losses of the diffusion process and the VAE is chosen to be one. Have you considered different relative factors? Could this improve the reconstruction?
5. Data representation:
1. You state that “Both the mass and energy are included [for the particle level information] in the representation to improve robustness to numerical issues.” Assuming that in the end these are not exactly compatible and px^2+py^2+pz^2+m^2 won’t equal E^2, which observable do you choose to present your results?
2. At a later point you state that no lepton mass is learned at all. How is this possible given your general choice of particle level observables?
6. Uncertainties and sampling:
You state “However in this application, the uncertainties obtained from sampling the model are strictly larger than the statistical uncertainties in the distributions.” Can you explain why this is the case? Have you checked the calibration of the distributions obtained from sampling multiple times for the same event? Do the migration matrices between reco and unfolded particle level observables reproduce the truth?
7. Deviations
You state “It is then unsurprising that the network tends to return the mean value of ην in events that are particularly difficult to unfold. “ While the argument seems logical it is surprising when looking at the plot. While there the unfolded distribution is overshooting at 0 one can observe a deficit directly next to it. Following the argument, the events that are particularly difficult to unfold would be right in the middle of the bulk. Can you expand on the argument?
Recommendation
Ask for minor revision
Report
The reply of the authors has addressed all comments I made before. I recommend the improved manuscript for publication in SciPost Physics.
Requested changes
There is just one minor thing I noticed that the authors might want to adjust: Towards the end of section 4.1, the variable $M$ does not refer to the previously used multiplicity at detector level, but rather the mass. Maybe they could use $m$ instead?
Recommendation
Publish (easily meets expectations and criteria for this Journal; among top 50%)
Author: Kevin Greif on 2025-01-27 [id 5153]
(in reply to Report 1 on 2024-12-03)
Many thanks for the support, and the helpful comments that we feel improved the paper!
In regards to the use of the variable M, we think from the context that it is clear that we are referring to the mass and not to the detector level multiplicity, so we've elected to leave this unchanged.
Sincerely,
The authors
Author: Kevin Greif on 2025-01-27 [id 5154]
(in reply to Report 2 on 2024-12-17)Errors in user-supplied markup (flagged; corrections coming soon)
Many thanks for these helpful and detailed comments! We've just submitted a new version which we hope addresses all of these concerns. A few responses to the above are below.
Sincerely,
The authors
===================================
1. Typos and similar
- "They have been been"
- "D_{DENOISE}, This"
- Eq. 10/11 the tilde is off.
- Fig.3 should include \bar{t}, \bar{b}etc.
- Fig. 5: Some labels (a,b,c) are only half visible.
- Fig .13 c seems to be missing the SM truth line?
- empty page 33
**Thank you for catching all of these mistakes. They have been corrected in the new version, with the exception of Fig. 13c where the SM unfolded and SM truth lines cover each other. You can see this if you look in the plot with the densities instead of the ratio pad.**
2. You state "Because generative approaches require only synthetic data during training, they do not suffer from this limitation." This is only half correct. Indeed the initial unfolding algorithm can be trained on unlimited statistics of the MC. Once you observe a prior dependence are you have to iterate, the limited amount of available data does become a limiting factor for the iteration. Maybe it is possible to argue that the unfolding is still less affected though.
**We completely agree with this point. To some extent iteration will spoil this nice property of generative methods, though the degree to which this will happen is unclear. This is a subtle detail, so rather than explain it at length in the text we have stated that generative methods might be less sensitive to the number of data events and added an explanatory footnote.**
3. Fig. 1: This figure is very hard to follow. While it becomes clearer when reading the paper, a few design choices could be altered to ensure better readability.
1. Fix a main direction that conveys the main task you want to illustrate.
2. Avoid edges when possible, even if the figure becomes bigger.
3. The execution of the unfolding should lead to an arrow from (x_0,.., x_N) to the particle decoder, which I would consider the main direction. However that direction is not indicated.
**We thank the reviewer for this insightful comment. We have rearranged the figure and added flow lines to demonstrate the data-flow during both training and inference. We have also re-oriented the figure to be read top-to-bottom and left-to-right.**
4. Related to Fig. 1 and Fig. 2 several choices are not clear:
1. The role of y0 as a learnable parameter. It seems to be independent from any input. So it is a learnable constant? Why is this not covered by the Multiplicity Predictor itself?
**Appending a learnable parameter before an attention block is a standard method to furnish fully permutation invariant predictions from transformer blocks. It is constant in the sense that it does not depend on any input as you note, but it is not constant in that the optimizer is free to change its value during training. We’ve added a reference to the paper that introduces “class attention”, which is essentially what we use here, for the curious reader.**
2. Is the sampling of the latent space in the VAE considered part of the encoder or the decoder and at which part does the output of the Denoising Network enter?
**Sampling from the VAE is achieved by computing the initial-time latent diffusion process: X(0) = alpha(0) * X + sigma(0) * eps where X is the output from the encoder and eps is a standard normal distribution sample. Unlike in a traditional VAE, the noise of the vae latent is determined by the (learned) noise schedule at time 0. We couple the VAE and diffusion like this because during inference the diffusion process ultimately produces X(0), not the original X. Therefore, our decoder must be capable of accurately decoding slightly noisy samples instead of the original encoded vectors. Details are provided in the third and fourth paragraphs of Section 3.1.**
3. Is the transformer encoder in Fig. 2 an encoder or the denoising network in Fig. 1?
**This is the particle denoising network. We have added more detail to the figure caption to aid in reading the figure’s message.**
4. The relative factor between the losses of the diffusion process and the VAE is chosen to be one. Have you considered different relative factors? Could this improve the reconstruction?
**It would be possible to introduce relative factors between the terms in the loss. However if you do this, you lose the interpretation of the loss function as a generalized ELBO loss as in Ref. 50. In other words, the choice of one is principled, and not just a guess.**
5. Data representation:
1. You state that “Both the mass and energy are included [for the particle level information] in the representation to improve robustness to numerical issues.” Assuming that in the end these are not exactly compatible and px^2+py^2+pz^2+m^2 won’t equal E^2, which observable do you choose to present your results?
**If we are presenting mass results, we use the mass prediction, and if we are presenting energy results, we use the energy prediction. For all derived observables like the top kinematics presented in Section 4.2, we use the energy predictions as inputs to the calculations. We do this because derived observable kinematics are essentially identical using both the predicted mass and energy components with the exception of the mass and energy themselves. Prior work focused on parton-level unfolding has also used a physics-informed constraint loss which tries to ensure that the predicted mass and energy match when derived from the other, but we do not include this constraint loss because the mass is much smoother for particle-level unfolding. **
2. At a later point you state that no lepton mass is learned at all. How is this possible given your general choice of particle level observables?
**The previous draft of the paper contained an unclear footnote which suggested the lepton masses are not learned. This is not the case, and the lepton masses are learned just like the jet masses. However since the true lepton masses are well known, there is no reason to unfold them in practice. For this reason we simply drop the lepton masses from the particle level event and replace them with their known values. The footnote in the paper has been updated to make this clear.**
6. Uncertainties and sampling:
You state “However in this application, the uncertainties obtained from sampling the model are strictly larger than the statistical uncertainties in the distributions.” Can you explain why this is the case? Have you checked the calibration of the distributions obtained from sampling multiple times for the same event? Do the migration matrices between reco and unfolded particle level observables reproduce the truth?
**This is the case because the two uncertainties arise from very different sources. We included this statement to make the point that the statistical uncertainty in the distributions can be ignored, since the uncertainty from sampling the model is much larger. The statistical uncertainty in all distributions is quite small since we have 1 million events in our testing set, which is used to draw all figures. The uncertainty from sampling the model arises from two sources. First there is the variance of the true posterior that arises from the fact that the detector response is not invertible. This uncertainty would exist even for a “perfect” unfolding method. Second, there is any additional variance that is added by the model, since the neural networks are not perfectly expressive. The first source of uncertainty is inherent to the problem, and the second source of uncertainty is a shortcoming of the method. We have not checked the calibration of the distributions or binned migration matrices, but we believe these checks are beyond the scope of this paper, especially considering the method fails to reproduce the 1 dimensional marginal distributions of some important observables like the top quark mass peaks. Recent pre-prints have included these checks for similar methods with good results.**
7. Deviations
You state “It is then unsurprising that the network tends to return the mean value of ην in events that are particularly difficult to unfold. “ While the argument seems logical it is surprising when looking at the plot. While there the unfolded distribution is overshooting at 0 one can observe a deficit directly next to it. Following the argument, the events that are particularly difficult to unfold would be right in the middle of the bulk. Can you expand on the argument?
**“Particularly difficult to unfold” was imprecise language. We have updated this sentence to clarify that the neutrino eta being underconstrained is different from an event being difficult to unfold due to large migrations produced by the detector. With this language update we hope the explanation makes more sense. The model returns the mean value of the distribution when the event configuration is such that the neutrino eta is not constrained. Apparently these events are often those that have neutrino eta close to 0. We believe an explanation for why this should be the case is beyond the scope of this paper, but suspect it can be understood by considering the W boson mass constraint typically used to assign a value for the neutrino eta in semi-leptonic ttbar events.**

---

## Round 2 · Author Response

Apologies for the delay in re-submission. Our lead author was away on a summer internship and has only recently returned. We have already responded to the reviewers in the comments section below. These comments refer to version 2 of the paper, now on the arXiv. We hope the new version addresses all of the reviewers concerns.
Sincerely,
Kevin for the team

---

## Round 2 · List of Changes

1. Section 4.2, 2nd paragraph: Addition of discussion of the ability to sample a single detector-level event multiple times.
2. Section 4.2, 7th paragraph: Discussion of corner plots presented in Appendix E
3. Section 6, 2nd paragraph: Remove sentence “This lack of prior dependence strongly motivates the use of VLD for unfolding”.
4. Section 6, 5th paragraph: Add statements on data and code availability.
5. Appendix B: Add definitions of the distance metrics used.
6. Appendix E: Add corner plots.

---

## Round 3 · Author Response

Thank you for the very helpful comments received. This is the second re-submission which we hope addresses all remaining concerns.
Sincerely,
The authors

---

## Round 3 · List of Changes

Introduction:
- Moderated claim that generative methods do not suffer from a limited number of data events, given application of an iterative method to eliminate prior dependence may be necessary.
Section 3:
- Added a reference to "class attention" to help the curious reader understand the learnable vector appended to the encoded detector level features.
Section 4:
- Clarified language in the discussion of the excess density predicted by the model at neutrino eta of 0.
In addition several typos have been fixed and the blank page in the appendices has been removed.

---

## Editorial Decision

editorial_decision: